# Ubiquitous macropinocytosis in anthozoans

**Philippe Ganot\*, Eric Tambutté, Natacha Caminiti-Segonds, Gaëlle Toullec[†], Denis Allemand, Sylvie Tambutté**

Marine Biology Department, Centre Scientifique de Monaco, Monaco, Monaco

**Abstract** Transport of fluids, molecules, nutrients or nanoparticles through coral tissues are poorly documented. Here, we followed the flow of various tracers from the external seawater to within the cells of all tissues in living animals. After entering the general coelenteric cavity, we show that nanoparticles disperse throughout the tissues via the paracellular pathway. Then, the ubiquitous entry gate to within the cells' cytoplasm is macropinocytosis. Most cells form large vesicles of 350–600 nm in diameter at their apical side, continuously internalizing their surrounding medium. Macropinocytosis was confirmed using specific inhibitors of PI3K and actin polymerization. Nanoparticle internalization dynamics is size dependent and differs between tissues. Furthermore, we reveal that macropinocytosis is likely a major endocytic pathway in other anthozoan species. The fact that nearly all cells of an animal are continuously soaking in the environment challenges many aspects of the classical physiology viewpoints acquired from the study of bilaterians.

## Introduction

Unicellular eukaryotes use endocytosis, in particular phagocytosis, to probe their surrounding medium and obtain their next meal. In higher metazoans, that is bilaterians, phagocytic cells mostly represent an essential branch of the immune system, whereas feeding is mediated by digestive organs which degrade complex molecules into nutrients further absorbed by individual cell (*Desjardins et al., 2005*; *Goodman, 2010*; *Buchon et al., 2014*; *Cosson and Soldati, 2008*). In early branching metazoans such as sponges or anthozoans (corals and sea anemones), comparatively little is known in regard to the cellular processes by which the surrounding medium with its nutritive, signaling or infectious contents, is processed.

From a histological and anatomical point of view (*Allemand et al., 2011*; *Tambutté et al., 2011*), anthozoans (included in the phylum Cnidaria) are simple organisms: their mouth is surrounded by a crown of tentacles and serves both for feeding and excreting waste, their body cavity hosts digestive and reproductive organs. The tissues are made up of two cell layers, the ectoderm and the endoderm separated by an acellular layer of mesoglea. Stony corals (scleractinians) represent the branch of anthozoans that synthetize a skeleton. They are colonial, meaning that the individuals called polyps are linked together by the cœnosarc, a portion of continuous tissue that connects polyps together (*Figure 1*). Their tissues are qualified as oral or aboral depending on whether their ectoderm is facing seawater or the skeleton, respectively. Between the oral and aboral tissues lies the common central cavity of the colony, called the coelenteric cavity. Most reef-building corals are symbiotic, hosting in their endodermal cells photosynthetic dinoflagellates (Symbiodiniaceae) from which they derive a large part of their organic nutrients (*Muscatine and Porter, 1977*). At the interface between the calcifying ectodermal layer (also called the calicoblastic epithelium) and the skeleton lies the extracellular calcifying medium (ECM) (*Tambutté et al., 2011*). Despite their relatively simple tissular organization, many questions remain concerning the biology/physiology of corals. Firstly, the process of cell to cell communication and exchange is unknown; secondly, these animals depend on both heterotrophic and autotrophic feeding but the cellular pathways involved in these two feeding

**\*For correspondence:**
pganot@centrescientifique.mc

**Present address:** [†]EPFL Lausanne, Lausanne, Switzerland

**Competing interests:** The authors declare that no competing interests exist.

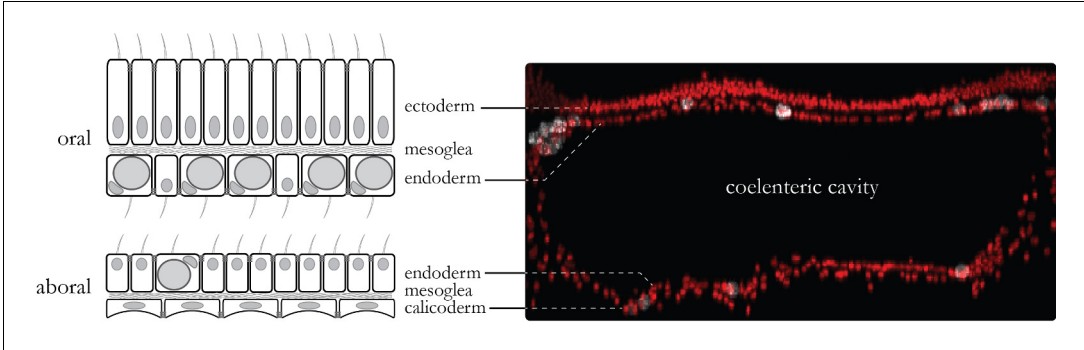

**Figure 1.** Coenosarc anatomy. The tissue connecting individual polyps is composed of two tissues. The oral tissue is composed of the oral ectoderm (in contact with the sea water) and the oral endoderm (which contains Symbiodiniaceae in most cells) separated by the acellular mesoglea. The aboral tissue is composed of the aboral endoderm (which contains Symbiodiniaceae in some cells) and the calicoblastic ectoderm (responsible for the formation of the skeleton) separated by the mesoglea. Tissue layers have their basal side contacting the mesoglea whereas motile cilia and septate junctions mark their apical side. In between the two endodermal layers lies the coelenteric cavity which carries most of the internal fluids.

modes are unknown; thirdly, these animals build a calcareous skeleton but the path by which ions and organic molecules are supplied to the site of calcification remains debated. Importantly, although cell-cell communication, feeding and calcification seem to refer to different aspects of biology, these processes all involve the properties of the coral's epithelial cell layers. Indeed, as is the case for any multicellular organism, the coral epithelial cell layers (i) serve as functional barriers, forming selectively permeable interfaces between compartments of different chemical composition and regulating the transport of ions and molecules, (ii) act as sensors of the extracellular environment, triggering the cellular response to extracellular variations and (iii) are involved in cell-cell communication.

Transport of ions/molecules from one cell layer to another cell layer and between the two tissues has been shown to occur through two different pathways: a paracellular and a transcellular pathway. The paracellular pathway (i.e. passing in between cells) occurs through septate junctions (*Davy et al., 2012*; *Tambutté et al., 2012*; *Barott et al., 2015*), the properties of which reflect the cell layer permeability. This paracellular pathway is selective and depends on the size and charge of each transported entity (*Tambutté et al., 2012*). Based on physiological experiments, it has been shown that the permeability to ions is different between the oral and aboral tissues (*Bénazet-Tambutté et al., 1996b*). Based on ultrastructural and molecular data, this could be due to constituent differences between the septate junctions of the oral and the aboral cell layers (*Ganot et al., 2015*). In regard to coral anatomy and fluid circulation, it must be noted that 1) only the oral ectoderm is covered by a layer of mucus which can affect the velocity of ion/molecule/fluid direct uptake from seawater; 2) both oral and aboral endoderms are bathed by the coelenteric fluid the renewal rate of which depends on mouth closure/aperture and on fluid diffusion through the paracellular pathway; 3) the calcifying aboral ectoderm is neither in direct contact with seawater nor with the coelenteric fluid. Therefore, both paracellular diffusion and internal fluid renewal should be considered when studying the transport of fluid/molecules/ions through coral epithelial cell layers. The supply of ions to the ECM involves a transcellular pathway with channel/carriers located on the apical and basal membranes of the polarized calicoblastic cells. For example, bicarbonate and calcium have been proposed to be transcellularly transported through these carriers/channels (*Barott et al., 2015*; *Zoccola et al., 2015*; *Zoccola et al., 2004*; *Zoccola et al., 1999*). Transcellular and paracellular transport can only help explain how ions and specific molecules are transported from seawater to the different cell layers and to the ECM (*Allemand et al., 2011*). In fact, molecules and fluids presumably also enter and exit coral cells through endocytosis and exocytosis, respectively. These processes have been poorly characterized in corals but some authors suggest that trans-cellular transport of vesicles occurs during the calcification process (*Mass et al., 2017*). In these models, it

has been proposed that seawater is endocytosed into vesicles and enriched in carbonate ions in the calcifying cells. These vesicles would then be transported and their content exocytosed in the ECM.

Endocytosis includes different cellular pathways which can be classified according to vesicle size. Clathrin- and caveolin- dependent/independent internalization are endocytic processes using small (less than 100 nm) vesicles (*Hansen and Nichols, 2009*; *McMahon and Boucrot, 2011*). On the other hand, endocytic processes operating via large vesicles (200 nm – 5 µm) are typical of phagocytosis and macropinocytosis (*Bloomfield and Kay, 2016*; *Kerr et al., 2009*; *Mayor and Pagano, 2007*). Macropinocytosis is a highly conserved process found from unicellular eukaryotes to metazoans, similar to that of phagocytosis, although direct contact with the internalized material is not required as for phagocytosis (*McNeil, 1981*; *Levin et al., 2015*; *Doherty and McMahon, 2009*). Macropinocytosis is an actin-driven process: plasma membrane ruffling and closing takes up surrounding extracellular fluids and its content into large vesicles, which may further fuse intracellularly with endosomal vesicles (*Swanson, 2008*). In humans, macropinocytosis can be constitutive or induced and has been involved in various cellular functions including antigen presentation, cell metabolism, as well as in cancer (increased nutrient uptake), pathogen entries (from viruses to protozoans), and therapeutics (drug entry gate) (*Bloomfield and Kay, 2016*; *Levin et al., 2015*; *Recouvreux and Commisso, 2017*; *Canton et al., 2016*; *Canton, 2018*; *Yoshida et al., 2018*; *Zhang et al., 2011*; *Lim and Gleeson, 2011*; *Commisso et al., 2013*). Hence, macropinocytosis can be considered as the endocytic pathway allowing non-specific endocytosis of large nanoparticles, up to 5 µm in diameter (*Swanson, 2008*).

In anthozoans (which include sea anemones, gorgonians and corals), macropinocytosis has never been characterized despite its potential role in fluid uptake. Of note, pioneering studies concerning the uptake of India ink (*Olano and Bigger, 2000*) or dissolved organic matter (DOM) by epithelial cell layers (*Apte et al., 1996*; *Schlichter, 1982*) might be worth being revisited with regard to more recent macropinocytosis data. In the present study, we used several type/size of nanoparticles, that is gold nanoparticles, latex beads and dextrans to characterize the transport of fluids and associated molecules/nanoparticles in the coral *Stylophora pistillata* for which many physiological, biochemical and molecular data are available (*Tambutté et al., 2011*). Using mostly confocal microscopy approaches, we followed nanoparticle movement from the surrounding seawater to the coelenteron, then to the different tissue layers via the paracellular pathway and further into the cells' cytoplasm. We show that the large majority of cells continuously take up large volumes of their surrounding medium using macropinocytosis. Macropinocytosis was confirmed both with transmission electronic microscopy and specific inhibitor experiments. In addition, we show that macropinocytosis is polarized from the apical to the basal side of cells in all tissues. Thus, the oral ectoderm facing the seawater directly absorbs the media, more precisely what is trapped in the mucus covering the animal; the two endoderms lining the coelenteron directly absorb the coelenteric fluid, whereas the calicoblastic ectoderm samples the ECM. In terms of dynamics, the mucus apparently represents a mesh slowing down large particles from being immediately taken up by the oral ectoderm. The coelenteric cavity is filled up within ca. 5 min, whereas only nanoparticles below 20 nm width reach the ECM with an additional ca. 10 min delay, likely due to the passage through the septate junctions filtering the paracellular diffusion to the ECM. Finally, we also described macropinocytosis in the sea anemone *Anemonia viridis* and in the octocorallian *Corallium rubrum*, thereby revealing macropinocytosis as a major endocytic process in anthozoans.

## Results

### Dextran uptake by *Stylophora pistillata* occurs through vesicles

To investigate the endocytic route in the coral *Stylophora pistillata*, we first analysed tracer molecule uptake within the whole organism down to the cell level by incubating microcolonies in sea water containing gold dextran-coated nanoparticles of two sizes (3 nm and 10 nm). Using transmission electron microscopy (*Figure 2*) we detected gold nanoparticles within large intracellular vesicles bigger than 200 nm in diameter, in both the ectodermal and the endodermal layers. Dextran particles were never observed free in the cytoplasm. This result shows that nanoparticles follow an endocytotic pathway with the aspect of vesicles varying from simple to multivesicular body endosomes. Such large vesicles could be characteristic of phagocytosis, macropinocytosis, or even late

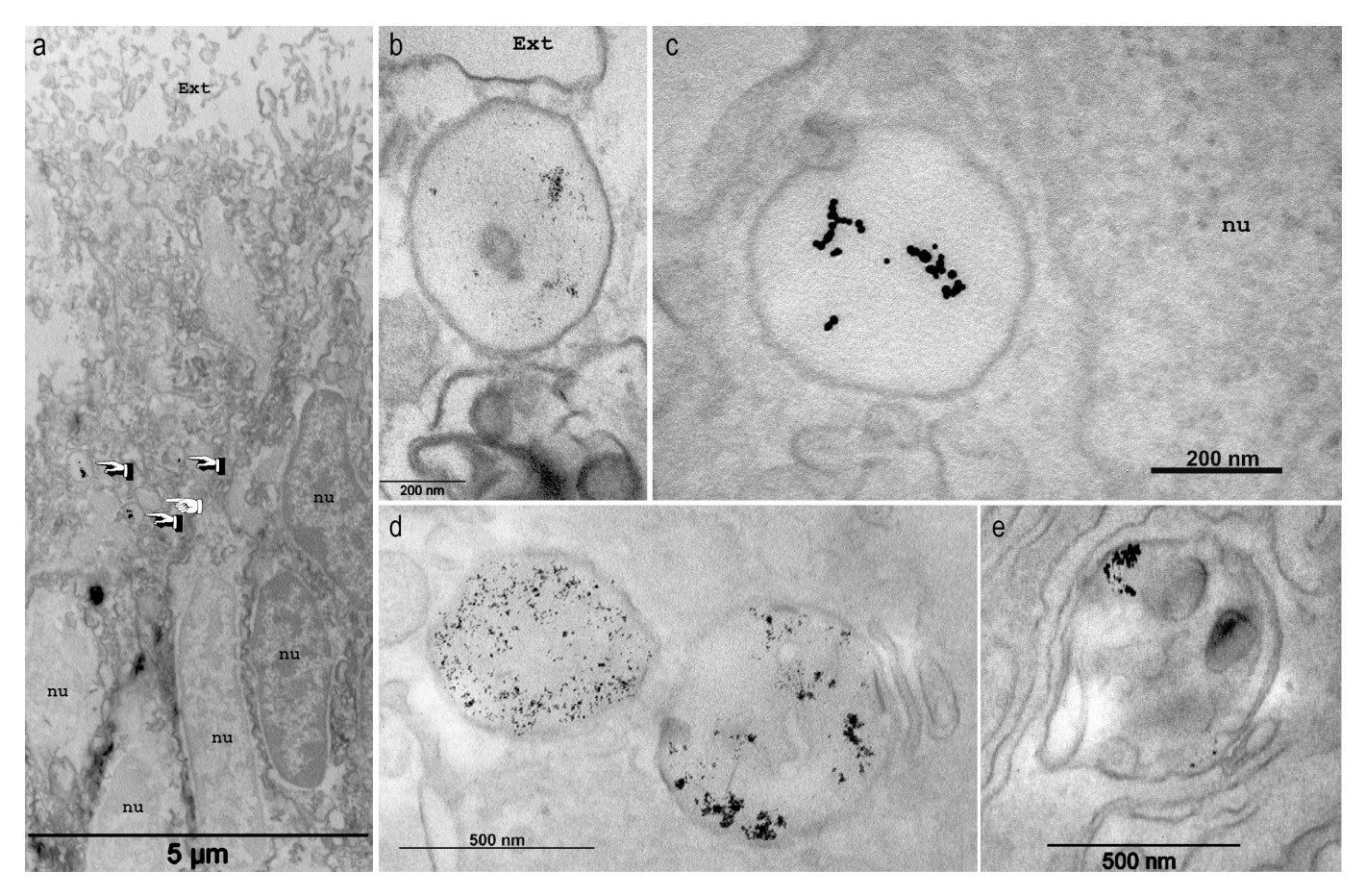

**Figure 2.** Endocytosis of gold dextran-coated nanoparticle by large vesicles. Microcolonies were incubated in sea water complemented with Gold dextran coated nanoparticles of 3 nm (**a,b,d,e**) or 10 nm (**c**) in diameter and imaged with transmission electron microscopy in the ectoderm (**a–b**) or the endoderm (**c–e**). Samples here were not contrasted with OsO$_4$. In (**a**) gold nanoparticles are detected in 3 out of the four vesicles (finger pointing) present in the tissue layer, but not in other parts of the cytoplasm. Higher magnifications (**b–e**) clearly delineate the large vesicles containing the nanoparticles. Of note, multivesicular body endosomes containing the nanoparticles (d, vesicle on the right, and e) were also observed. *Ext*, external sea water medium; *nu*, nucleus.

The online version of this article includes the following figure supplement(s) for figure 2:

**Figure supplement 1.** Macropinosomes formation.

endosomes resulting from the intracellular fusion of smaller early endosomes. However, when osmium post-fixation was performed (*Figure 2—figure supplement 1*), electron micrographs showed large vesicle formation right below the apical membrane of both ectodermal and endodermal cells which is typical of macropinocytotic processes.

## Dynamics of dextran uptake differs between cell layers and dextran size

To get further insight into coral endocytosis, we next sought to qualitatively characterize fluid/molecule dynamics within whole *S. pistillata* animals using common fluorescent dextran as endocytic markers (*Kerr et al., 2009*; *Clarke et al., 2002*; *Wang et al., 2014*; *Li et al., 2015*; *Chen et al., 2018*). Furthermore, we asked whether dextran uptake by the different cell layers was particle size dependent. Therefore, we performed a time course experiment using two sizes of dextrans, 3 kDa and 10 kDa (D3K and D10K) (*Figure 3* and *Figure 3—figure supplements 1–3*). The control at T0 shows no fluorescent labeling (ie. no autofluorescence) whatever the epithelial layer analyzed. After 5 min, dextran molecules were visible at the apical surface of all epithelial layers except for the

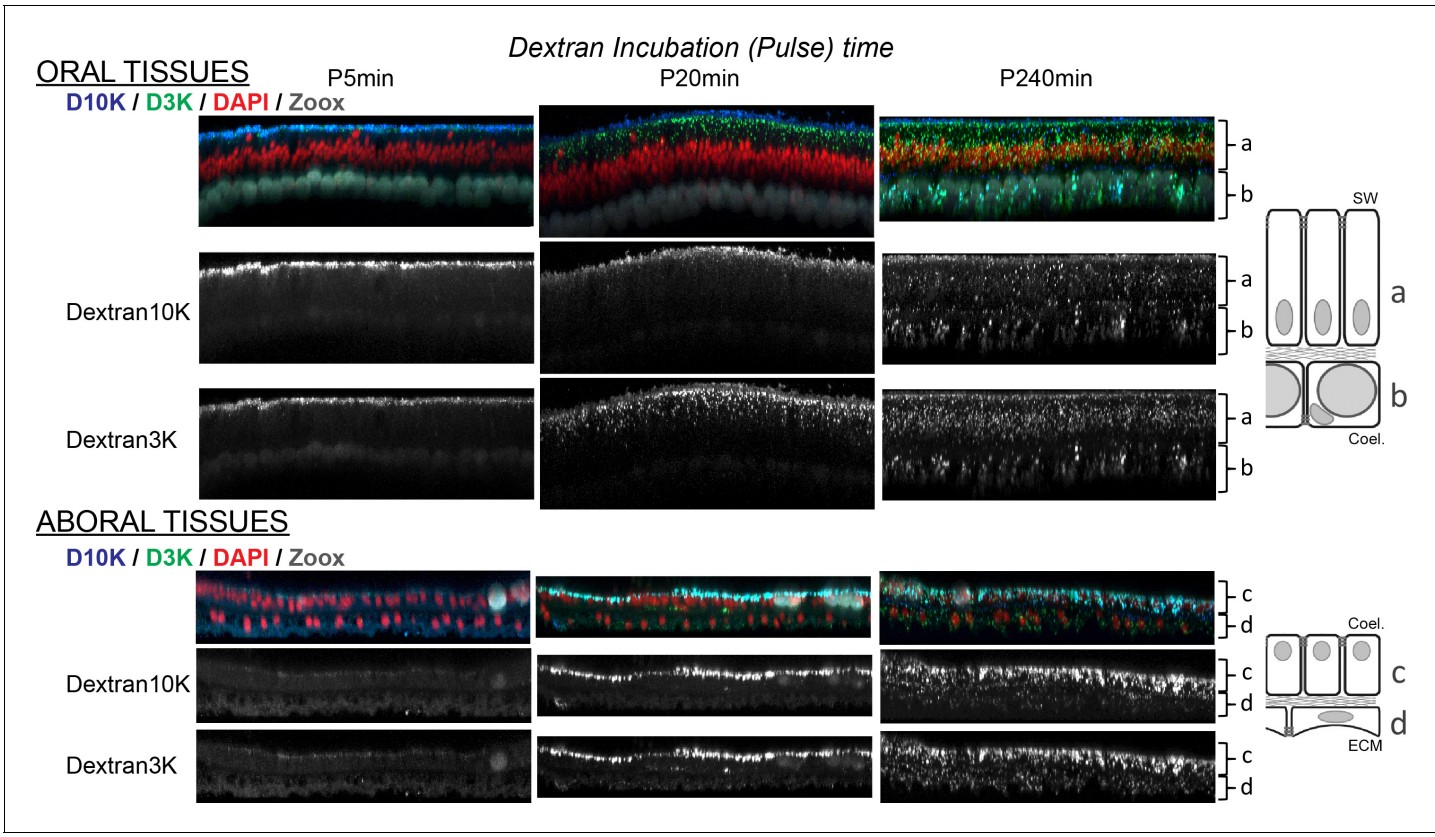

**Figure 3.** Kinetic of dextran uptake by Stylophora pistillata. Incremental time length of Dextran 3K and 10K incubation shows gradual penetration of dextrans inside the tissues. Note that this figure only shows 5, 20, and 240 min' incubation times, see *Figure 3—figure supplement 3* for complementary time points. All images correspond to y-projections of Z stacks acquired through the oral (top panel) and the aboral tissues (bottom panel). The different tissue layers, depicted besides the photos as a,b,c,d, correspond to the oral ectoderm, the oral endoderm, the aboral endoderm, and the calicoblastic ectoderm respectively (SW: sea water; coel.: coelenteron; ECM: Extracellular Calcifying Medium). Dextran 3K and 10K individual channel acquisitions are shown in black and white, merged are shown in color: Dextran 10 KDa (D10K) in blue, Dextran 3 KDa (D3K) in green, DAPI (nuclei) in red and chlorophyll autofluorescence from the Symbiodiniaceae present in the endodermal tissue layers in gray. Each photograph represents 144 μm wide and 9 μm depth tissues.

The online version of this article includes the following figure supplement(s) for figure 3:

**Figure supplement 1.** Sample processing.
**Figure supplement 2.** Imaging strategy.
**Figure supplement 3.** Incremental pulse (0, 5, 10, 20, 60, 240 min incubations) lengths of Dextran 3K and 10K show the Dextrans' gradual penetration inside the tissues.
**Figure supplement 4.** Probable macropinocytosis in Corallium rubrum.

aboral calicoblastic ectoderm where the labeling appeared 10–20 min later. With incremental pulse duration (from 10 to 240 min incubations), dextran labeling increasingly invaded every cell layer in an apparent apical to basal manner. Importantly, the pattern of dextran uptake in the oral ectoderm appeared to be size-dependent, with preferential internalization of D3K versus D10K. This was not the case within both endoderms where dextran fluorescence in the cells was similar whatever the size of the dextran (*Video 1*). Of note, D3K and D10K appeared to co-localize only in a few vesicles (*Figure 3*). However, close inspection of the separate dextran confocal emission channels revealed that both dextrans co-localized in most vesicles in all cell layers, albeit with seemingly variable relative concentrations (ratio). Altogether, this data suggests that the dynamics of molecule internalization is tissue specific and size-dependent and suggests that internalization occurs at the apical membrane in all tissues. To test whether the uptake of dextatran into large vesicles could be extend to other distant anthozoans, we incubated the octocorallian *Corallium rubrum* in a similar manner (*Figure 3—figure supplement 4*). Apparently all cells showed D3K and D10K uptake into large vesicles.

## Dextran vesicle progression inside the cells

Next, we further characterized vesicular intracellular progression in all *S. pistillata* tissues through a series of pulse-chase experiments using three dextran sizes (3, 10 and 70 kDa). From the previous time course experiment, we determined that 15 min was the time necessary to visualize dextran within cells of all *S. pistillata* epithelia. Indeed, after 15 min incubation, most D3K and D10K signals were, as expected, concentrated at the apical side of the different cell layers (*Figure 4A*). However, after a 4 hr chase period preceded by a wash (ie. dextran was no longer present in the incubation medium), the dextran signal was more centrally positioned inside the cells (compare with nuclei position), confirming an apical to basal progression in all tissues. Again (see above), in the oral epithelium, penetration of the D10K was delayed as compared with D3K (*Figure 4B*). This was even more obvious when comparing 3 KDa and 70 KDa dextrans (*Figure 4C*). These results indicate that endocytic vesicles of dextran follow an apical to basal progression within all *S. pistillata* tissues reminiscent of endocytic processes.

## Endocytosis occurs at the apical side of the cells

In order to firmly conclude that the uptake of dextran occurs apically, we performed double dextran pulse experiments (*Figure 5*). After a first 15 min pulse and a 4 hr chase, another 15 min pulse was applied using only D3K so that dextran size did not come into play. Combinations of different fluorochromes were used in pulse 1 and 2 to make sure that the choice of the fluorochrome had no effect on the results, which it did not (not shown). Whichever the epithelial cell layer considered, we show that after the pulse1-chase-pulse2 experiment, dextran fluorescence from the first pulse was

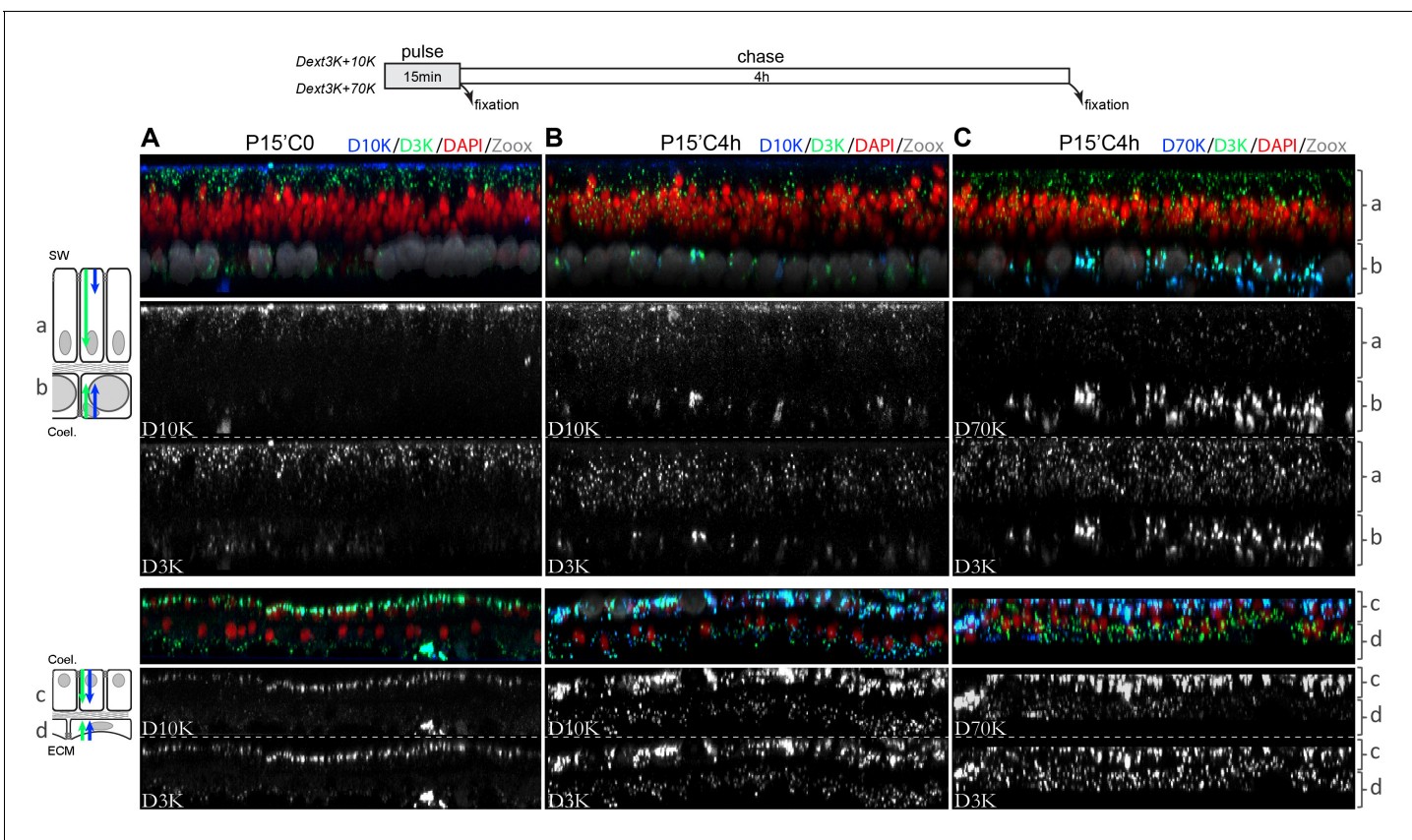

**Figure 4.** Cellular progression of the dextran uptake. Dextran pulse of 15 min followed by 4 hr chase. Microcolonies were incubated for 15 min in sea water supplemented with D3K and D10K and fixed (A), or pulsed with D3K and D10K (B) or D3K and D70K (C) and left for an additional 4 hr in sea water before fixation. Tissues and labelling legends are as in *Figure 2*. Note that in all tissues, dextrans appear to follow an apical to basal progression inside the cells. Contrary to the other cell layers, the oral ectoderm have preferential endocytosis toward smaller dextran molecules (D3K >> D10K>D70K). Each photograph represents 144 µm wide and 9 µm depth tissues.

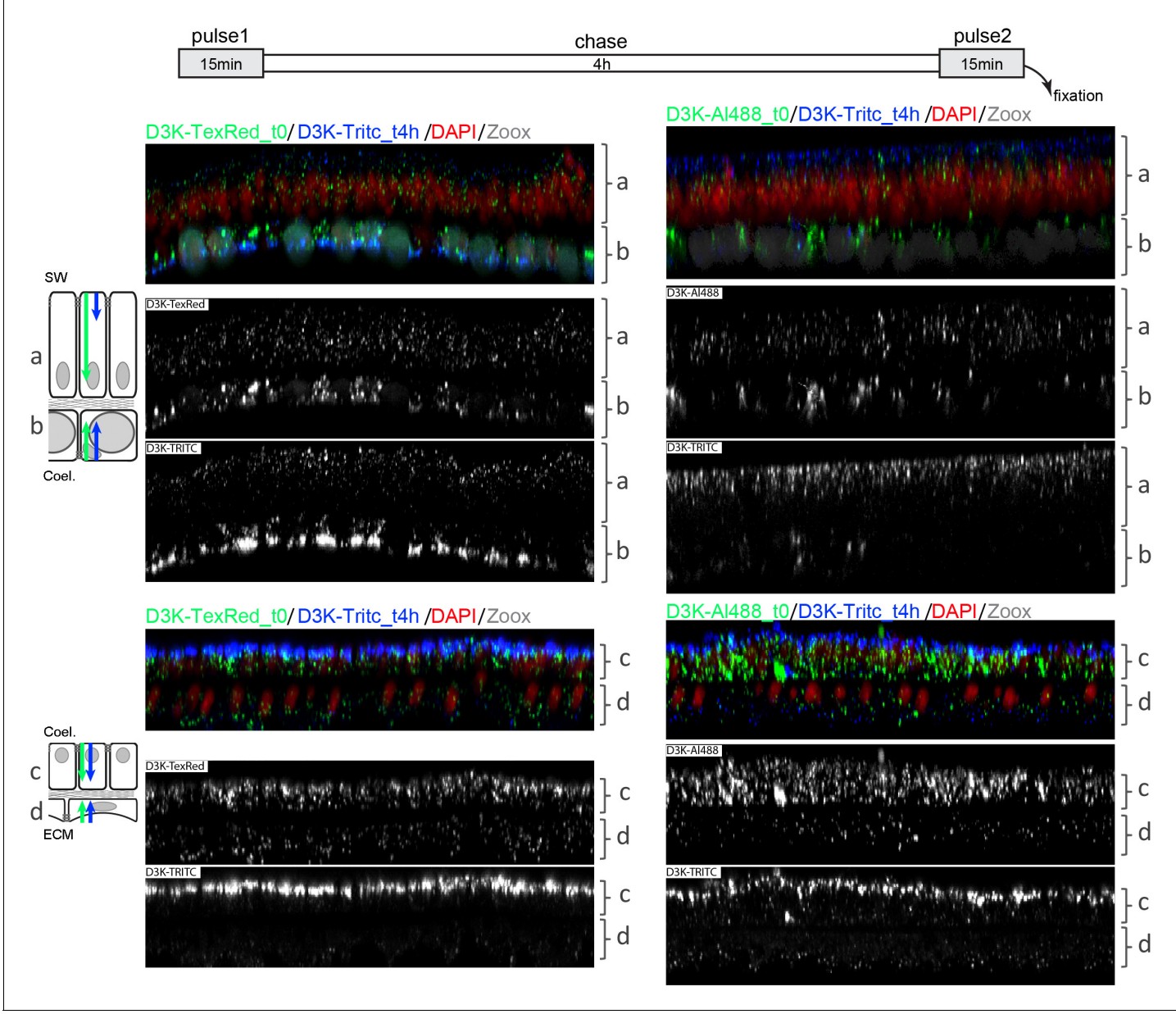

**Figure 5.** Apical to basal endocytosis: two 15 min pulses separated by a 4 hr chase. Microcolonies were first incubated for 15 min with D3K either coupled with Texas-red (left panel) or Alexa 488 (right panel), chased in sea water for 4 hr to let the dextran progress through the epithelial cells' cytoplasms, and then incubated for another 15 min with D3K coupled with TRITC. Regardless of the cell layer, the second pulsed dextran was always found at the apical face of the tissues whereas the first pulsed dextran was visible throughout the cells' cytoplasms. Each photograph represents 144 μm wide and 9 μm depth tissues.

The online version of this article includes the following figure supplement(s) for figure 5:

**Figure supplement 1.** Dextran uptake from the Coelenteron of the sea anemone Anemonia viridis tentacles.

**Figure supplement 2.** Dextran uptake from the Coelenteron of the sea anemone.

observable within the cells whereas the fluorescence of the dextran from the second pulse was only found at the apical side of the cells. Combined with the results of *Figure 4*, these results clearly demonstrate that dextran endocytosis occurs on the apical side of all cell layers.

In the case of the endoderms, the results accordingly show that the up-taken fluids/molecules came from the coelenteron and not from fluids absorbed basally from the mesoglea, as the apical sides of both cell layers face the coelenteron. This was corroborated with experiments performed on

the sea anemone *Anemonia viridis* whereby tentacles were filled with D3K and D10K (*Figure 5—figure supplement 1*). In this latter set-up, only the endodermal cells (and not the ectodermal cells) were bathed on their apical sides with the dextran solution. Similar to what is observed in corals, dextran uptake gradually increased in the endodermal cells from one to 15 min (*Figure 5—figure supplement 2*), implying that endodermal cells apically endocytose dextrans from the coelenteron.

In corals, since the apical side of the oral ectoderm lines seawater and since the apical side of the aboral calicoblastic ectoderm lines the extracellular calcifying medium (ECM), we reasoned that dextran uptake by these two layers had to come from the seawater and the ECM, respectively (note that seawater is separated from the apical side of the oral ectoderm by the mucus layer). Therefore, apical endocytosis of dextran by the calicoblastic ectoderm implies that dextran first follows the paracellular pathway from the coelenteron to the ECM.

## Paracellular diffusion through the aboral cell layers

To verify Dextran paracellular diffusion from the coelenteron to the ECM, we recorded movement of fluorescent dextran D3K between the cells composing the aboral tissue with live imaging of microcolonies grown on coverslips at the edge of the microcolony (where there are gaps in between the growing crystals). As was already observed with the tracer calcein in *Tambutté et al. (2012)*, dextran diffuses along the paracellular pathway through septate junctions (*Video 2*). Septate junctions are 15–20 nm in width and control the perm-selectivity of molecules. Therefore, in order to further characterize the paracellular diffusion through the septate junctions, we followed the flow of latex beads of 20 and 200 nm in diameter (particles ten to hundreds of times bigger than dextrans; *Table 1*, *Figure 6*). Confocal observation of the different tissues showed clear intra-cellular localization of the beads in the oral ectoderm as well as the oral and aboral endoderms, regardless of their size. However, in the aboral calicoblastic ectoderm, no particle uptake was detected. This result clearly shows that the oral ectoderm and the two endoderms can absorb the beads (and not only dextran-based molecules) from the seawater and the coelenteron, respectively. However, barriers, likely represented by the septate junctions of the two aboral cell layers (plus potentially the mesoglea), prevent the paracellular diffusion of latex beads over 20 nm in diameter towards the ECM and thus, prevent their access to the calicoblastic ectoderm.

Taken together, the data with dextran, gold nanoparticles and latex beads show that particles of different sizes and nature enter the cells into large vesicles analogous to the macropinosomes described in other taxa. Particles are

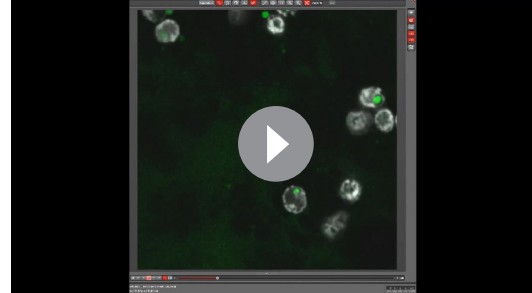

**Video 2.** Paracellular diffusion. The video correspond to screencasts of the LasAM program graphical interface (Leica) displaying the time-laps acquisition (xzyt) in the video mode. The laterally grown microcolony was set in an incubation chamber and analyzed by inverted confocal microscopy from beneath, at the edge of the microcolony where there are gaps in-between the growing crystals. Time laps imaging was recorded (single Z section, 5sec/image). After 3 min 30 s of recording, texasRed-D3K was added to the medium. D3K is in green (TexasRed detected 615–625 nm); Symbiodiniaceae (10 µm diameter) are in gray. Note the timer display at the bottom right corner of the screen. Within tens of seconds, the paracellular labeling is apparent.

https://elifesciences.org/articles/50022#video2

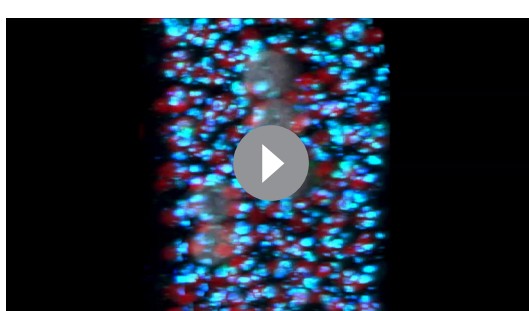

**Video 1.** Macropinosomes. A branch of *Stylophora pistillata* Coral was incubated for 30 min with D3K and D10K, then fixed. The video shows a 3D reconstruction (LAS-AF) of a Z-stack encompassing the aboral endoderm. Symbiodiniaceae (10 µm diameter) are in gray, nuclei (3 µm diameter) in red, texasRed-D3K in green and TRITC-D10K in blue. Note the macropinosomes labeled with D3K and D10K which appear as merged color.

https://elifesciences.org/articles/50022#video1

**Table 1.** Tracers' endocytosis.

| Tracer | Size (nm) | Charge* | Oral ecto. | Oral endo. | Aboral endo. | Calico | Para-cellular |
|---|---|---|---|---|---|---|---|
| dextran 3 KDa Alexa488 | 1.3 | (-) | ++ | ++ | ++ | ++ | + |
| dextran 3 KDa TexRed | 1.3 | (+/-) | ++ | ++ | ++ | ++ | + |
| dextran 10 KDa TRITC | 2.3 | (-) | + | ++ | ++ | ++ | + |
| dextran 70 KDa TRITC | 6 | (-) | - | ++ | ++ | ++ | + |
| gold-dextan-coated-3nm | 3 | (?) | + | + | + | na | na |
| gold-dextan-coated-10nm | 10 | (?) | - | + | + | na | na |
| microsphere-20nm | 20 | (–) | + | + | + | - | - |
| microsphere-200nm | 200 | (–) | + | + | + | – | - |

*charge: (?) unknown, (+/-) zwitterionic; (-) anionic; (–) highly negatively charged.

engulfed from the medium lining the apical side of the cells. With regard to the calicoblastic ectoderm, although small molecules such as dextran can pass through the paracellular pathway, latex beads of bigger sizes (see *Table 1* for particle sizes) are blocked by the septate junctions.

## The size of dextran vesicles is in the 350–1500 nm range

In order to further characterize the endocytotic vesicles in the coral *S. pistillata*, we performed a quantitative analysis (size and number) of dextran labeling in all epithelial layers at two time points, after a pulse of 15 min or after a 15 min pulse/4 hr chase. Representative images of dextran labeling within the different cell layers are shown in *Figure 7A*. As aforementioned, the absence of color merging reflects the ratio variability in terms of dextran concentrations within vesicles, whichever the tissue. In the oral ectoderm, most vesicles were 350 to 800 nm in width, with or without the chase period (*Figure 7B*). As previously observed, the number of vesicles was greater with smaller dextran. In the oral endoderm, D3K was rapidly endocytosed (P15C0), the diameter of the first observable vesicles being in the 350–800 nm range. However, after 4 hr of chasing, the size distribution shifted towards much larger vesicles (up to 1.5 µm in width), which possibly implies vesicle fusion with time and/or cytoplasm progression. The same holds true for the aboral endoderm except that the number of vesicles was significantly higher (maybe due to lower steric hindrance since Symbiodiniaceae are much less numerous than in the oral endoderm) and that there was no preference for dextran size at 15 min pulse. Finally, in the calicoblastic aboral ectoderm, only a few vesicles were counted after the 15 min pulse, likely because of the delay inherent to the passage via the paracellular pathway (D3K passing faster than D10K). After 4 hr of chasing, the vesicle number was higher although vesicle size remained in the 350–800 nm diameter range regardless of dextran particle size. Hence, vesicle quantification reflects what was observed previously, i.e. 1) the vesicles in the ectodermal cells are smaller than those in the endodermal cells (350–800 nm *versus* 350–1500 nm), potentially due to vesicular fusion; 2) the oral ectoderm shows dextran uptake size dependence; 3) the calicoblastic cells have a delayed kinetic of dextran uptake as compared to the other cell layers.

## The endocytotic pathway of dextran is macropinocytosis

Combining all the data described above, we hypothesize that the endocytic pathway described here in *S.pistillata* is macropinocytosis. Macropinocytosis has been shown to be specifically inhibited by EIPA (that blocks the NHE exchanger), by PI3K (Phosphoinositide 3-kinases) inhibitors such as wortmannin (*Clague et al., 1995*; *Araki et al., 1996*), and by actin inhibitors such as latrunculin (*Cosson et al., 1989*; *Koivusalo et al., 2010*; *Gold et al., 2010*). This is the case in mammalian cells as well as in the protist *Dictyostelium discoideum* (*Neuhaus et al., 2002*; *Williams and Kay, 2018*), implying highly conserved cellular pathways among Eukaryota. To ascertain that macropinocytosis is indeed the endocytic pathway observed in corals, we monitored the uptake of dextran in the presence of these inhibitors. Contrary to the DMSO control experiment (*Figure 8A*), incubation with EIPA efficiently impaired the formation of macropinosomes (*Figure 8B*) in all cell layers. When incubated with 1 µM wortmannin, dextran uptake was impaired both quantitatively and qualitatively (*Figure 8C*) in every cell layer except for the aboral epitheliums. Indeed, Dextran (D3K) uptake in the

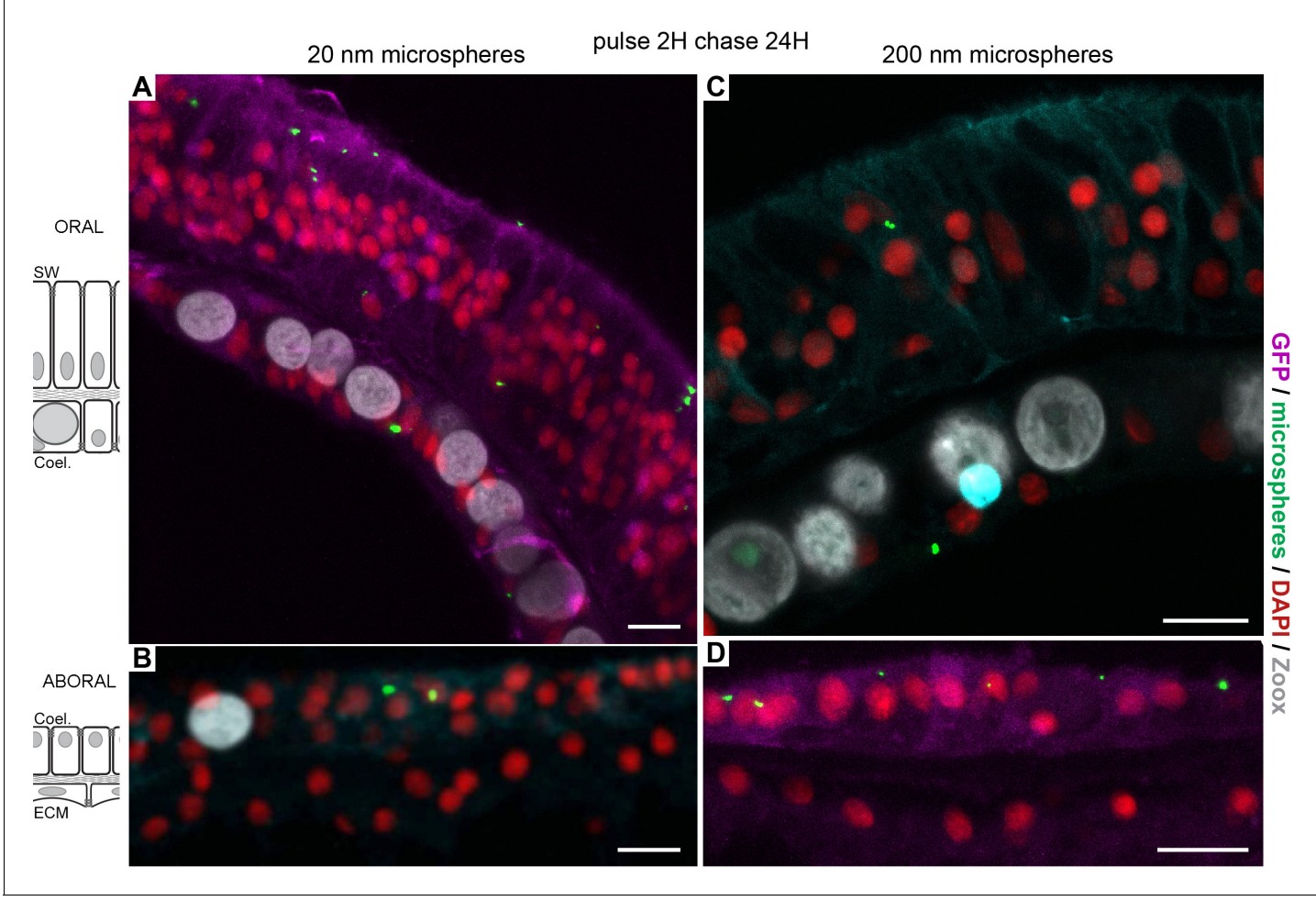

**Figure 6.** Latex beads paracellular diffusion and macropinocytosis. microcolonies were incubated with 20 nm and 200 nm latex beads in order to challenge (i) the paracellular barrier and (ii) dextran specific endocytosis. Incubations of Fluorescent latex beads for 2 hr followed by a chase of 1 day show that the three tissue layers (A, oral ectoderm; B, oral endoderm; C, aboral endoderm; D, calicoblastic ectoderm) in contact with the medium could engulf particles of 20 nm (**A,B**) and 200 nm (**C,D**) in diameter. Thus, after passing via the polyps' mouths (data not shown), beads enter the coelenteric cavity and are endocytosed by the endoderm tissue layers, although with less efficiency than dextran molecules. Alternatively, they are directly acquired from the sea water by the oral ectoderm. However, in the calicoblastic cells, neither 20 nm nor 200 nm beads were detected. The paracellular barrier made by septate junctions prevented the access to the extra calcifying medium, and thus the calicoblastic cells. Scale bar = 10 μm.

aboral ectoderm was faint and in the aboral endoderm, the signal was stretched or horse-shoe like, and did not resemble the typical large and round vesicles seen in the control experiments.

Vesicle formation was also inhibited by Latrunculin (LatA) and inhibition was reversible (*Figure 9*). Similar to the previous experiments, dextran uptake was inhibited in the two oral cell layers as well as in the calicoblastic ectoderm in the presence of 500 nM LatA. In the aboral endoderm, although endocytosis was modestly compromised at 500 nM (*Figure 9A*), it was completely abolished at 1000 nM LatA (*Figure 9—figure supplement 1*). F-Actin inhibition by latrunculin has been shown to be reversible in vertebrate cultured cells (*Spector et al., 1983*). To evidence such reversibility in corals, after the inhibitory step previously described (step1), only half of the colonies was fixed while the other half was set back into regular seawater for 4 hr in order to washout the LatA. Then these (semi-)colonies were assayed for dextran uptake. After this second step, endocytosis was restored in all tissues treated with LatA (500 nM and 1000 nM), except for the calicoblastic ectoderm after the 1000 nM LatA treatment as integrity of this particular tissue was compromised at such a concentration. Thus, vesicle formation in *S. pistillata* was sensitive to EIPA, Wortmannin, and LatA, corroborating our previous results suggesting that vesicle formation follows the macropinocytosis route.

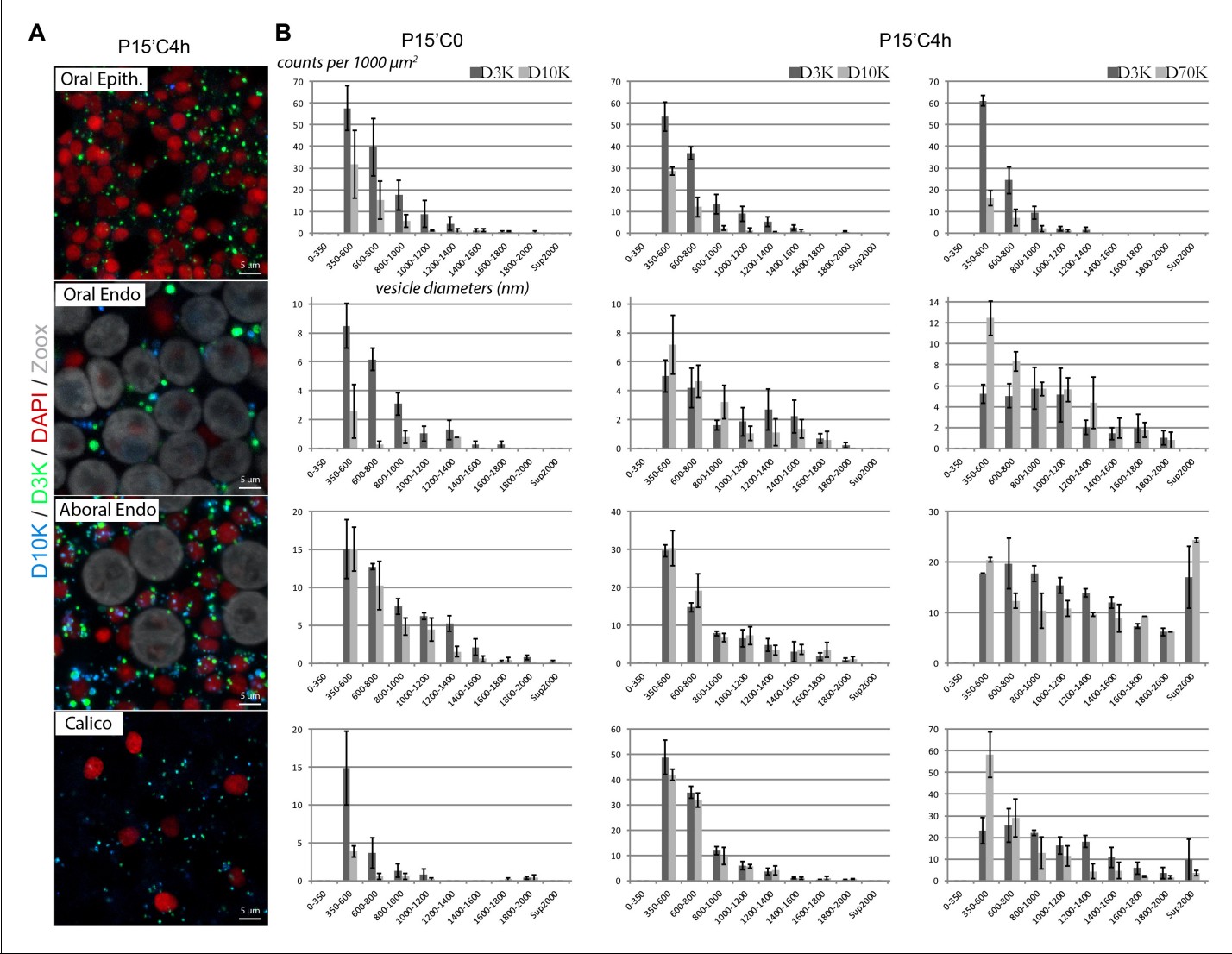

**Figure 7.** Quantitative analysis of the dextran endocytic vesicles. High magnification Z stack sections encompassing the different individual tissue layers (from top to bottom: oral ectoderm, oral endoderm, aboral endoderm, calicoblastic ectoderm) were acquired after the same pulse-chase experiments than in *Figure 3A*) Representative Z projections for each tissue after D3K/D10K 15 min pulse/4 hr chase (legends as in *Figure 3*). Note the variable relative content of the co-localizing dextrans from one vesicle to another. (**B**) For each experiment, the diameters (nm) of the dextran vesicles were measured then ranked according to their size distribution and their counts were normalized per square units (detailed in Appendix 1). Note that the very large vesicles (D > 2 μm) observed in the aboral ectoderm with D3K-D70K were in most cases due to the piling of smaller vesicles that were recognized by the imageJ program as one single artefactual vesicle after Z-stack projection.

Finally, in order to get further insight into the mechanism of vesicle formation, we next analyzed the cellular F-actin network during macropinosome formation. To do so, we had to minimize the time of decalcification prior to phalloidin labeling. Hence, we used microcolonies grown on cover-slips, which were only partially decalcified before Phalloidin staining. We also used a SP8 confocal equipped with hybrid detectors to increase the detection sensitivity. In this experiment, only the aboral tissues (ie. those in contact with the coverslip) could be visualized (*Figure 10*, *Videos 3*–*5*). After incubation with D3K for 15 min, we clearly observed an F-actin dent, that we termed 'cap', associated with the forming macropinosomes at the apical side of the endodermal and calicoblastic cells. Importantly, these actin caps did not surround the dextran vesicles, at least in the initial stage of formation. Our interpretation of the 3D reconstruction image stacks is that the formation of an intracellular actin cap could create a depression at the surface of the plasma membrane provoking the engulfment of the extracellular liquid.

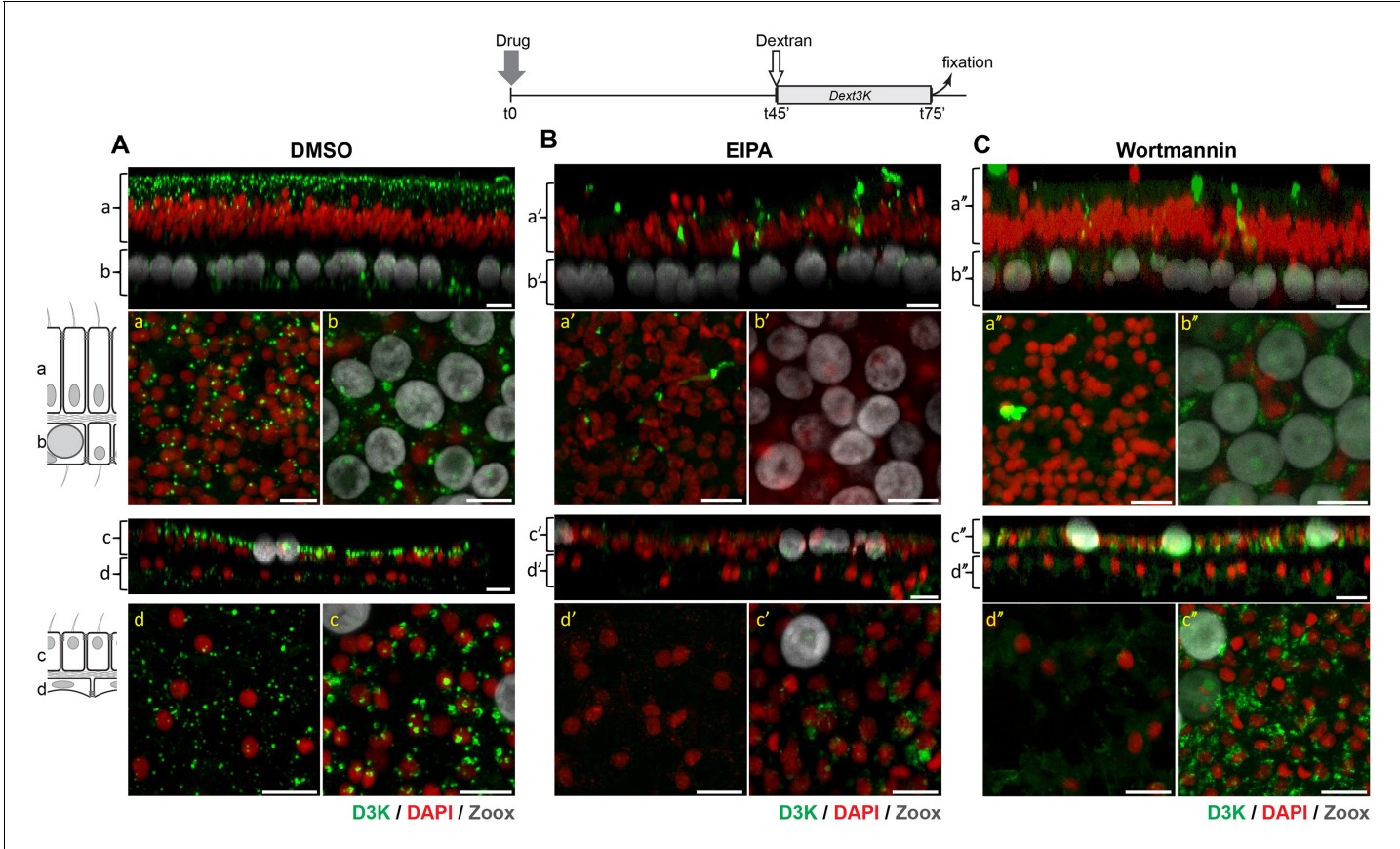

**Figure 8.** EIPA and Wortmannin inhibits macropinocytosis in corals. Microcolonies were first incubated in the presence of DMSO (control) (A), 100 µM EIPA (B) or 1 µM Wortmannin (C) for 45 min. Then D3K was added to the incubations and pulsed for an additional 30 min before being fixed and processed for confocal analysis. Top (a,b, a',b') and bottom (c,d,c',d') panels corresponds to oral and aboral tissue analyses, respectively. In each panel, the top picture is a y projection of a Z-stack through the tissue layers as in *Figure 3*, and the two bottom pictures correspond to a Z projection of a stack embracing individual tissue layers at higher magnification as in *Figure 7A*. Although DMSO has no effect on dextran endocytosis, EIPA and Wortmannin strongly impaired dextran internalization. Note that for the Wortmannin experiment, PMTs were increased to pick up the low signals of dextran intake. Scale bars = 10 µm.

These results combined with the results concerning vesicle size observed during uptake of dextrans and beads and the localisation of vesicle formation observed at the apical membrane of the cells clearly show that the endocytotic process observed here is macropinocytosis and that these vesicles can be named macropinosomes.

Our data suggest that macropinocytosis is ubiquitous to most cells since every field we observed with the microscope showed a labeling pattern similar to that shown in *Video 1*. Such a process is not specific to the scleractinian coral *Stylophora pistillata* since we also observed dextran internalisation in the sea anemone *Anemonia viridis* and in the red coral, *Corallium rubrum*. Macropinocytosis thus appears as an ubiquitous process found in different anthozoan species.

## Discussion

Here, we show that macropinocytosis is a ubiquitous endocytic pathway found in all the cell layers of the coral *S. pistillata* (Hexacorallia/Scleractinia). Furthermore, we show that this pathway is conserved in anthozoans since we also observed similar vesicles in the sea-anemone *Anemonia viridis* (Hexacorallia/Actiniaria) and the red coral *Corallium rubrum* (Octocorallia/Scleraxonia); Hexacorallia and Octocorallia are the two major subclasses of Anthozoa, which likely diverged more than 600 Mya (*Kayal et al., 2018*; *Guzman et al., 2018*). Such a ubiquitous use of macropinocytosis was largely unexpected as this process is restricted to specialized cells in the sister group of Bilateria.

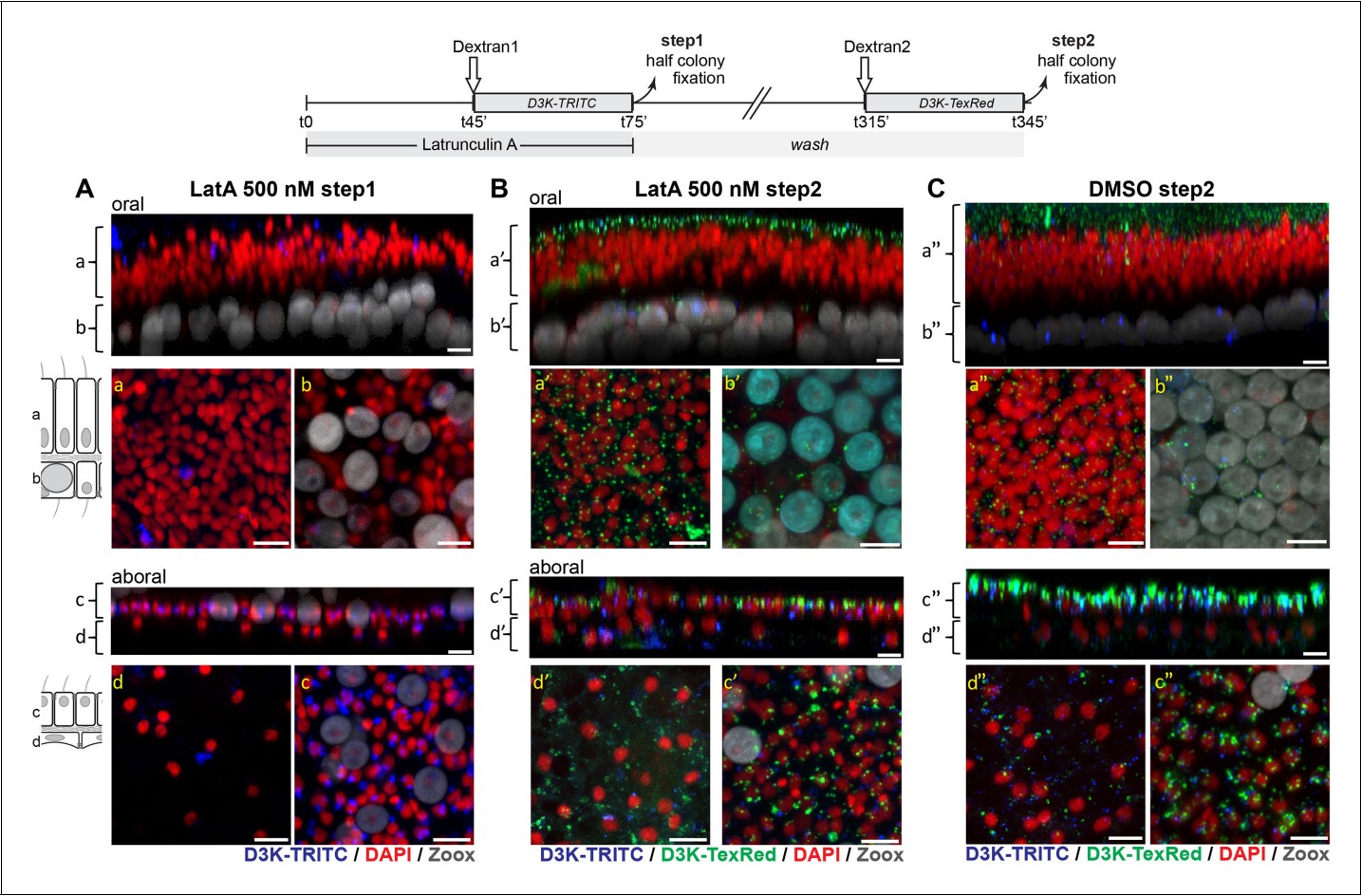

**Figure 9.** Actin is required for coral macropinocytosis: at T = 0, microcolonies were incubated in the presence of 500 (panels **A** and **B**) Latrunculin or control DMSO (panel **C**). At T = 45 min., D3K-TRITC was added to the incubation. At T = 75 min. colonies were cut into two halves, one half was fixed (step1) the other half was set to normal seawater. At T = 315 min., the second halves of the colonies were incubated with D3K-TexRed for another 30 min. before fixation (step2). Panels and legends are as in *Figure 7*. At 500 nM, micropinocytosis (intake of D3K-TRITC) is clearly inhibited in the oral and calicoblastic cell layers whereas only slightly impaired in the aboral endoderm (step1, Panel A) (note that macropinocytosis in the aboral endoderm is completely inhibitied at 1000 nM LatA, *Figure 9—figure supplement 1*). However, after 4 hr washing, inhibition was removed as macropinocytic activity (intake of D3K-TexRed) was restored (PanelB). Scale bars = 10 µm.

The online version of this article includes the following figure supplement(s) for figure 9:

**Figure supplement 1.** Latrunculin A (LatA) at 1000 reversibly inhibits macropinocytosis in the aboral endoderm.

## Characterization of macropinocytosis in anthozoans

Macropinocytosis is one of the endocytotic pathways that exists in eukaryotic cells, along with others such as clathrin-mediated endocytosis or phagocytosis (for more details see reviews *Kerr et al., 2009*; *Canton, 2018*; *Lim and Gleeson, 2011*). A budding structure from the plasma membrane is a prerequisite for any endocytotic pathway. In the case of macropinocytosis, it usually occurs from highly ruffled regions of the plasma membrane (flat sheet-like protrusions; *Swanson, 2008*) that extend from the cell surface, constrict and close, with membrane fusion producing a large intracellular vesicle containing a droplet of medium (*Doherty and McMahon, 2009*). Using transmission electron microscopy, we could not show outward extensions of the plasma membrane. However, we could clearly picture the vesicles as part of the apical plasma membrane, with apparent subsequent internalization of the extracellular medium, which is typical of macropinocytosis (*Figure 2*). Macropinocytosis requires a dynamic actin skeleton essential for contractile motility. In corals, the forming vesicles were associated with what appears as an inward actin 'cap' (*Figure 10* and *Videos 3–5*). Furthermore, dextran uptake was reversibly inhibited by latrunculin, a potent actin inhibitor (*Figure 9*).

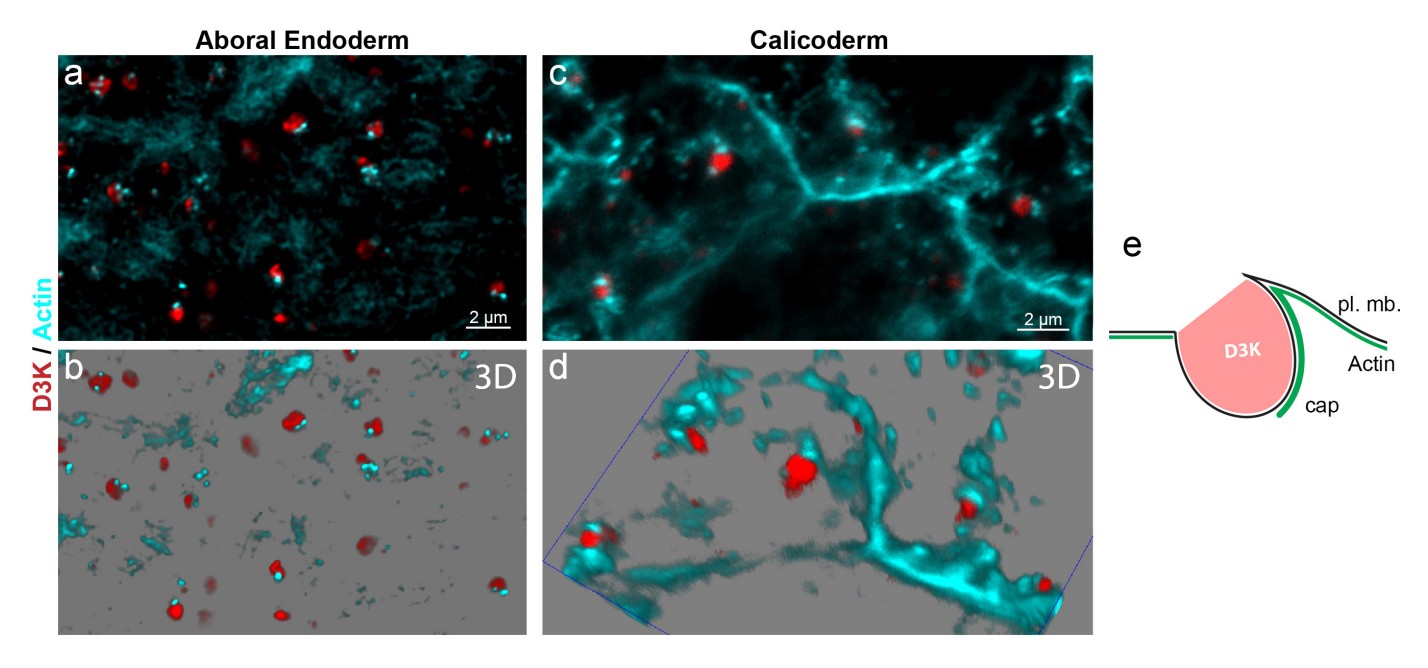

**Figure 10.** F-actin is linked to macropinosome formation. A microcolony grown on coverslip was incubated for 15 min. with D3K, fixed and labeled with phalloidin. Representative confocal Z-stacks (**a, c**) and 3D reconstruction snapshots (**b, d**) of the aboral endodermal (**a, b**) and calicoblastic (**c, d**) cell layers are shown; Phalloidin (F-actin) in cyan and D3K in red (see also *Videos 3–5*). For the vast majority of the forming macropinosomes, as visualized by D3K labeling, an intracellular 'cap' of actin is associated (**e**), illustrating the participation of F-Actin to macropinosome formation.

Macropinocytosis differs from other types of endocytosis by its unique susceptibility to Na$^+$/H$^+$ exchanger inhibitors (*Cosson et al., 1989*; *Koivusalo et al., 2010*). Here we show that EIPA, a specific blocker of this exchanger blocks macropinosome formation. Macropinocytosis is also a Phosphoinositide 3-kinase (PI3K) dependent process in eukaryotes (*Araki et al., 1996*; *Williams and Kay, 2018*), and blocking PI3K with wortmannin inhibited macropinocytosis (*Figure 8*). Thus, although the initial shape of the macropinosome-forming membrane may differ in corals versus vertebrate cells or *Dictyostelium*, the pharmacological approach used here suggests that the molecular processes leading to macropinocytosis are likely to be conserved throughout evolution. On the other hand, although macropinocytosis strongly supports our

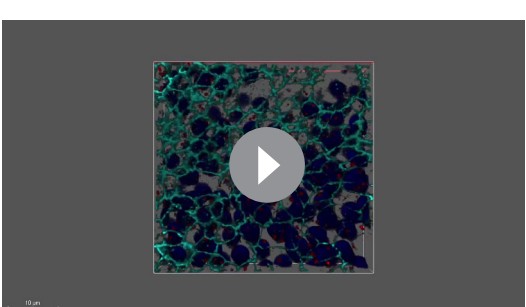

**Video 3.** Actin and macropinosomes, large view. A laterally grown microcolony was incubated in Dextran for 15 min, fixed and shortly decalcified before F-actin labeling with phalloidin. The video corresponds to a 3D reconstruction (LAS-X) of a Z-stack encompassing the aboral endoderm. The video describes a large view of the cellular F-actin network and the process of macropinocytosis (D3K uptake) in the endoderm. TexasRed-D3K in red (macropinosomes), F-actin in cyan and nuclei in blue.
https://elifesciences.org/articles/50022#video3

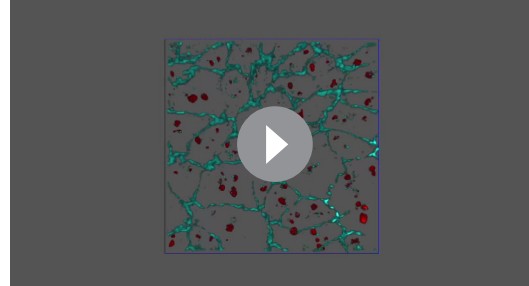

**Video 4.** Actin and macropinosomes, zoom into the endoderm. Same as in *Video 3* (endoderm), showing a closer view, without nuclei. Note the cap of actin (cyan) always associated with the forming macropinosome (red).
https://elifesciences.org/articles/50022#video4

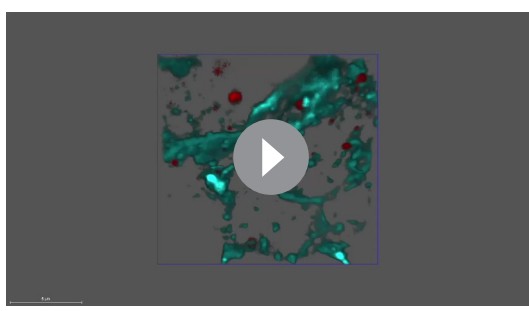

**Video 5.** Actin and macropinosomes, zoom into the calicoblastic cells. The video corresponds to a 3D reconstruction (LAS-X) of a Z-stack encompassing 3 cells of the calicoblastic ectoderm. Like for the endoderm, note the cap of actin (cyan) always associated with the forming macropinosome (red).
https://elifesciences.org/articles/50022#video5

observations of nanoparticle uptake, we cannot rule out the existence of other types of Clathrin-independent endocytosis (CIE) (*Ferreira and Boucrot, 2018*). Among CIE, the CLIC/GEEC (Clathrin-independent carrier/GPI anchored protein enriched endosomal compartments) pathway rapidly uptakes bulk fluid into early endosomes (*Howes et al., 2010*). The CLIC/GEEC pathway is driven by actin and is PI3K-dependent like macropinocytosis (*Hemalatha et al., 2016*), however it forms small tubular intermediates that we did not observe.

The size of the vesicles that we observed with the fluorescent fluid phase marker dextran is between 350 and 1500 nm, which is typical of macropinocytosis (*Levin et al., 2015*; *Swanson, 2008*). This large vesicle size, orders of magnitude larger than the molecules they capture, enables the uptake of solutes that are excluded from other endocytotic processes. Studies in mammalian cells have shown that macropinocytosis can be induced in a variety of cell types by growth factors and other stimuli whereas dendritic cells and macrophages perform macropinocytosis constitutively (*Canton et al., 2016*; *Canton, 2018*). Currently, from our observations, we cannot conclude whether macropinocytosis in corals is induced or constitutive and no data is available in the literature. However, we never performed an experiment without observing macropinocytosis in all cells, suggesting that the process occurs at all times.

## Particle diffusion and cell progression of macropinosomes

In corals, particles from the seawater medium are ingested into the coelenteric cavity via the mouth and can diffuse between the cells via the paracellular pathway (gated by 20 nm wide septate junctions; *Tambutté et al., 2012*; *Figure 11*). Our time course study shows that within 5 min, particles are detectable in the coelenteric cavity (*Figure 4*). Then to reach the calicoblastic layer, particles follow the paracellular pathway down to the ECM through the endodermal septate junctions, the mesoglea and the calicoblastic septate junctions, separating the coelenteron from the ECM. On the other side of the animal, particles can reach the oral ectodermal cells directly from the seawater medium. However, in this particular tissue layer, we observed a size dependent particle uptake. The oral ectoderm is covered by mucus which forms a diffusion barrier at the apical side of the cells (*Brown and Bythell, 2005*). This mucus layer is a gel-forming layer composed of polysaccharides that are constantly released by all coral species (*Bythell and Wild, 2011*). One possibility to explain the preferential endocytosis of smaller particles is that bigger particles would be trapped or delayed by this mucus layer. This is supported by the Transmission Electron Microscope observation of the content of the ectodermal versus endodermal macropinosomes (TEM observations): the former contain a heterogeneous medium with a fibrillary network (mucus-like) and the latter a homogeneous medium (*Figure 2—figure supplement 1*). Alternatively, other micropinocytic processes could ensure fluid phase uptake in addition to macropinocytosis. In the slime mold *Dyctiostelium*, concurrently to macropinocytosis, clathrin-mediated endocytosis is also able to capture Dextran fluorescent tracers into small vesicles that further fuse into large late endosomes (*Neuhaus et al., 2002*). Multiple endocytotic pathways presumably exist in anthozoans. Endocytosis of small size dextrans (e.g. D3K vs D70K) in small vesicles which would later fuse as larger endosomes may account for some selective aspects of fluid phase uptake. However, the endocytotic process in the different tissue layers is rapid. In *A. viridis*, macropinosome formation in the aboral endoderm takes 1 to 4 min. In *S. pistillata*, given the time necessary for the dextran to reach the cellular apical membrane, whichever the cell layer we can estimate to less than 5 min the formation of macropinosomes. Hence, given the kinetics of vesicle formation as well as their size and number, macropinocytosis likely accounts for most of the fluid phase uptake in coral tissues, including the oral ectoderm.

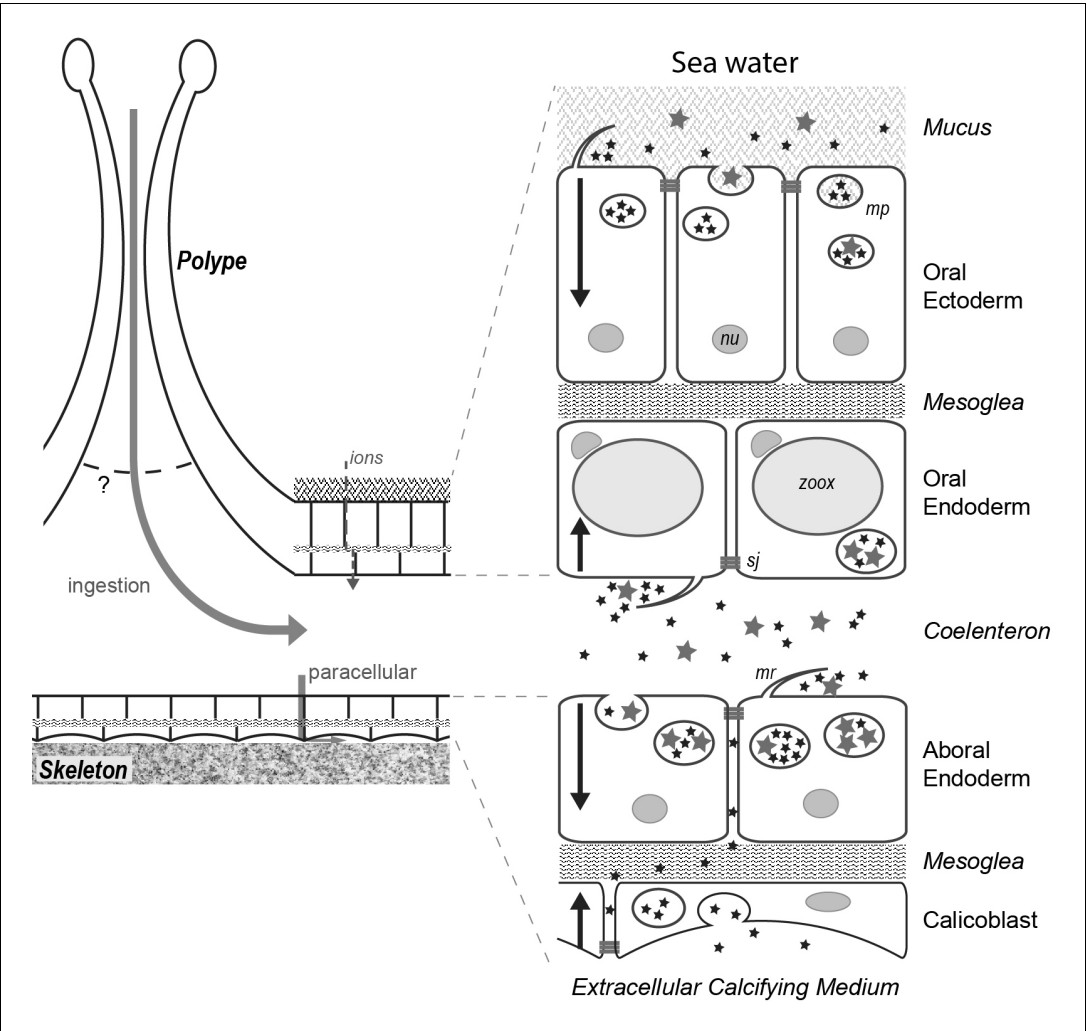

**Figure 11.** Fluids/particles flow from the sea water to the cell cytoplasm. Schematic representation of the movement of particles in the different tissue layers of a coral (see text). Stars represent particles of two different sizes (inferior and superior to 20 nm); mp, macropinosome; mr, membrane ruffling; nu; nucleus; sj, septate junction (apical side); zoox, Symbiodiniaceae.

Macropinocytosis occurs in apparently the vast majority of coral cells. Particles are engulfed into large vesicles originating from the apical membrane of the cells. These macropinosomes then move towards the inner part of the cells, down to the basal side. Macropinosomes in the endodermal cells are notably bigger than those of the ectodermal cells. A likely explanation is that in the endodermal cells, macropinosomes fuse during their progression. In vertebrate models, macropinosome fate after internalization varies depending on the cell type, as they can be recycled to the cell membrane or they can adopt degradative properties by fusing with lysosomes and undergoing a lysosome-dependent acidification (*Recouvreux and Commisso, 2017*). Further investigation would be required to decipher the fate of macropinosomes after internalization in anthozoans.

## Macropinocytosis and coral physiology

Macropinocytosis has general implications for coral physiology. With regard to feeding, symbiotic corals not only feed on autotrophic nutrition (symbiosis with Symbiodiniaceae) but also rely on two modes of heterotrophic nutrition, predation and assimilation of DOM (*Houlbrèque and Ferrier-Pagès, 2009*). The observation that in corals, diverse particles ranging in size from 1 to 200 nm are engulfed by most cell types suggests that exogenous particles such as DOM or particulate organic matter may enter these cells by macropinocytosis. The oral ectodermal cell layer was shown to

directly absorb DOM sources from the surrounding sea water by its apical membrane (*Schlichter, 1982*; *Sorokin, 1973*), which corresponds to the side of macropinocytosis, and thus corroborates the fact that macropinocytosis could be the DOM feeding pathway in the oral ectoderm. Yet, nutrition in corals involves the production of photosynthates by the Symbiodiniceae which is mostly hosted in the oral endodermal cells. How these photosynthates are further transported to the other tissues remains largely unknown. Bulk fluid absorption via macropinocytocis could form part of the mechanism for the uptake of photosynthates released in the coelenteric cavity. Hence, individual coral cells, and more generally anthozoan cells, would have the capacity to use macropinocytosis for both auto-and heterotrophic feeding.

With regard to calcification, recently Mass and collaborators observed mineral particles within the calicoblastic cells, with sizes varying from 400 nm to 9.4 µm in diameter, 400 nm being the most frequent (*Mass et al., 2017*). There was no direct evidence that these particles were localized inside vesicles but by analogy between the size of the mineral particles and the vesicles (380 nm) observed by *Clode and Marshall (2002)*, they deduced that the particles and the vesicles referred to the same object. In the present study, sizes of the macropinosomes in the calicoblastic cells were in the same size range as in *Clode and Marshall (2002)*, that is 350 to 800 nm. However, we cannot conclude whether the macropinosomes that we observed correspond to the same particles observed by *Mass et al. (2017)* and whether or not they contain mineral matter. Of major interest, apical to basal macropinocytosis is commonly observed during endocytotic events including micropinocytosis (*Shivas et al., 2010*; *Apodaca, 2001*) and is the prevalent endocytic pathway in all coral tissues, including calicoblastic cells. Thus, calicoblastic cells capture the extracellular medium that is present at the interface between the cells and the skeleton (i.e. the ECM). Taking this into account and the studies showing that the composition of the ECM is different from seawater (higher pH, Ca and carbonate [*Sevilgen et al., 2019*; *Cai et al., 2016*]), it is clear that 'calicoblastic' macropinosomes do not capture seawater from the medium but rather modified seawater. This brings novel insight into the recent model of calcification/biomineralization that implies transcytosis across the different cell layers where vesicular transport of ions occurs from the surrounding sea water down to the site of calcification (*Mass et al., 2017*). Of note, we never observed dextran labeled vesicles within the mesoglea, suggesting that vesicle passage from one tissue layer to the other does not occur. Overall, the role of macropinocytosis in the calicoblastic cells and its potential link with calcification awaits deeper investigations.

Finally with regard to corals and environmental stressors (*Hoegh-Guldberg et al., 2007*; *Hughes et al., 2017*), pollution by plastics is of primary concern for the global marine environment. Plastics do not dissolve in seawater but gradually fragment into microparticulate matter due to continuous physical, mechanical and chemical attack. With time, microsized plastics become nanosized particles that accumulate in the marine environment (*Cole et al., 2011*; *Law, 2017*). In the present study, we show that latex beads as large as 200 nm are able to enter the general coelenteric cavity, likely via the polyps' mouth, as we observed their accumulation at the oral opening (not shown). Then, they are found within the cells, although large latex beads were apparently internalized with less efficiency than smaller nanoparticles such as dextrans (see *Figures 4* and *6*). Thus, it seems likely that the size of plastic fragments (micrometer *vs* nanometer) also affects their internalization efficiency both at the coelenteric entry level and within the cells. However, current studies on corals have so far only addressed the toxicological impact of micrometer sized (entry inside the organism) and not nanometer sized (entry into the cells) plastics (*Reichert et al., 2018*; *Allen et al., 2017*; *Hankins et al., 2018*; *Rotjan et al., 2019*), as it has been done in other organisms (e.g. *Della Torre et al., 2014*). Since macropinocytosis allows internalization of nanoparticles of different sorts in all coral cells, this endocytic pathway should now certainly be taken into account in future eco-toxicology studies.

## Materials and methods

### Culture of corals and microcolonies

*Stylophora pistillata* mother colonies were grown in the culture facilities at the Centre Scientifique de Monaco where aquaria were supplied with flowing seawater from the Mediterranean sea (exchange rate 2% h$^{-1}$), at a salinity of 38, under an irradiance of 175 µmol photons m$^{-2}$ s$^{-1}$ of

photosynthetically active radiation (PAR) (400–700 nm) on a 12 hr light: dark cycle. Microcolonies were prepared from mother colonies by sectioning samples of 2–3 cm at the apex of a branch. Microcolonies were attached to a monofilament thread in the culture tanks and were used only after complete healing (3 weeks) (*Tambutté et al., 2012*; *Al-Moghrabi et al., 1993*). In the case of phalloidin staining and live imaging, another type of microcolony was used that grew laterally on coverslips (*Tambutté et al., 2012*). These types of microcolonies allow the direct visualization of the aboral tissues (*Tambutté et al., 2012*). *Anemonia viridis* sea anemone and *Corallium rubrum* red coral specimens were collected in Monaco and Marseille and grown in the culture facilities at the Centre Scientifique de Monaco as in *Bénazet-Tambutté et al. (1996a)*; *Le Goff et al. (2017)*. Corals were fed daily with frozen rotifers and twice weekly with live *Artemia salina* nauplii.

## Incubation of microcolonies with nanoparticles

Incubations of the coral *Stylophora pistillata* were all performed under similar conditions as cultured corals and experimental time always started between 13:00 PM and 14:00PM. Typical incubation experiments were carried out in 50 ml beakers with a magnetic stirrer. Incubations were all performed in sterile-filtered seawater (SFSW) freshly supplemented with 1 mg/L of lyophilized Rotifer (Ocean nutrition) per liter and further filtrated at 0.2 microns (supplemented sterile filtered seawater = S SFSW). Of note, initial experiments with dextran incubations of less than 60 min in SFSW alone gave similar results, but we preferred supplemented SFSW to avoid potential nutritional stress. Samples were incubated in 20 ml S-SFSW containing the nanoparticle for the desired pulse time and then optionally transferred to a new S-SFSW for the desired chase time, before fixation and further processing. Parallel control experiments without nanoparticles were always carried out. The time of the different pulses and chases are given in the text for each experiment.

In another experiment, a branch of the Mediterranean red coral *Corallium rubrum* was incubated with D3K and D10K for 4 hr. The branch was then fixed and thin slices of tissues were dissected, then mounted on a microscope slide.

## Nanoparticles used

### Fluorescent dextrans

Five type of dextran conjugates have been used (ThermoFisher D-34682, D-3328, D-3308, D-1817, D-1818). D3K (Dextran-Alexa488, 3000 MW, Anionic, Ex/Em = 495/519), D3K (Dextran-Texas Red, 3000 MW, lysine fixable, Zwitterionic, Ex/Em = 595/615), D3K (Dextran- tetramethylrhodamine, 3000 MW, lysine fixable, anionic, Ex/Em = 555/580), D10K (Dextran-tetramethylrhodamine, 10,000 MW, lysine fixable, Anionic, Ex/Em = 555/580), D70K (Dextran, tetramethylrhodamine, 70,000 MW, lysine fixable, Anionic, Ex/Em = 555/580) stock solutions were prepared from powder resuspended in SFSW at 10, 10, 10, 25 and 25 mg/ml, respectively, centrifuged at 12000 g, aliquoted and stored frozen at −20°C. D3K, D10K, and D70K working concentrations were 100, 250, and 500 ng/ml, respectively, to keep equivalent molarity.

### Fluospheres

Fluorescent latex beads of 20 and 200 nm diameter (ThermoFisher F-8887, red fluorescent carboxylate-modified Microspheres, Ex/Em = 580/605) were used at 2% final concentration in S-SFSW.

### Gold dextran-coated nanoparticles

5 ml of 3 nm or 10 nm Dextran coated Gold Nanoparticles (0.1 mg/ml H2O, Interchim Uptima 1P6270 and 1P3660) were complemented with 5 ml 2X artificial sea water (ASW, see below) and added to 120 ml S-SFSW in a beaker. Microcolonies were incubated in this solution for 6 hr before being processed for transmission electron microscopy.

### Transmission electronic microscopy of gold dextran-coated nanoparticles containing vesicles

Sample preparation and electron micrographs obtained with a JEOL transmission microscope were described in *Tambutté et al. (2007)*, except that no Osmium was added to avoid artifact contrast due to precipitation with aldehyde residues. Image contrast and brightness were adjusted with the Photoshop levels tool.

## Inhibitor experiments

Latrunculin A (Sigma), EIPA (5-(N-Ethyl-N-isopropyl) amiloride (Sigma), and Wortmannin (Selleck-chem), were dissolved in DMSO at 5 mM, 100 mM, and 10 mM, stock solutions, respectively, and diluted in S-SFSW to 500 nM (or 1000 nM), 100 µM [*Laurent et al., 2014*], and 1 µM, working concentrations, respectively. A control experiment with 1 µl/ml DMSO was carried out in parallel. For EIPA, and Wortmannin inhibitor experiments, microcolonies were first incubated in S-SFSW supplemented with the inhibitor for 45 min to allow the drug to act, and then transferred into S-SFSW supplemented with the inhibitor and Dextrans (D3K-TRITC) for 30 min, before fixation. For the Latrunculin experiment, microcolonies were also pre-incubated in the presence of the drug (500 nM, 1000 nM, or control DMSO) for 45 min before addition of D3K-TRITC and further incubated for 30 min. At the end of this first step, microcolonies were cut into two equal parts, one half being fixed and the other half being set back to regular seawater for 4 hr (washing). Then these second halves were subjected to another 30 min dextran labeling in S-SFSW with D3K-TexRed before fixation (step2), to assess the reversibility of the inhibition.

## General preparation of tissue sections for confocal microscopy

The procedure for sample fixation, decalcification, microdissection and labeling was previously described in *Ganot et al. (2015)*. Briefly (*Figure 1*), each sample was fixed in 50 ml chilled artificial-sea-water/paraformaldehyde (PAF) fixation buffer [425 mM NaCl, 9 mM KCl, 9.3 mM CaCl2, 25.5 mM MgSO4, 23 mM MgCl2, 2 mM NaHCO3, 100 mM HEPES pH = 7.9, 4.5% PAF] for 2 hr on ice. The skeleton part of the samples were decalcified in 50 ml [100 mM HEPES pH = 7.9, 500 mM NaCl, 250 mM EDTA pH = 8.0, 0.4% PAF, 0.1% Tween 20] at 4°C until skeleton complete dissolution (3–5 days). The remaining tissues were transferred into PBS for dissection under a binocular where polyps were removed using micro-scissors (Vannas), leaving only the cœnosarc (inter-polyp tissues). A/ For all dextran experiments, the dissected cœnosarc (~5×5 mm) was cut into two equivalent pieces, rinsed 3 times 10 min in PBS; with DAPI in the second rinse for nuclear counter-staining. Then, the two cœnosarc pieces were mounted, one with the calicoblastic (aboral) ectoderm up and the other one with the oral ectoderm up, in anti-fading medium (Slow Fade Gold antifade reagent, Molecular Probes) between microscope slide and coverslip separated by a frame layer (Gene Frame Thermo scientific) to compensate for the thickness of the cœnosarc. B/ For FluoSphere experiments, sagittal slices as thin as possible were manually cut using a scalpel through the cœnosarc. Slices were rinsed, counterstained with DAPI and mounted in anti-fading medium (Slow Fade Gold antifade reagent, Molecular Probes) between microscope slide and coverslip.

## Imaging of fluorescent nanoparticles on fixed tissue sections

Confocal imaging was performed using TCS SP5 DMI 6000 CS (Leica Microsystems) monitored by LasAF software platform with an HC PL APO 40x/1.3 oil CS2 objective. DAPI, Symbiodiniaceae, Green Fluorescent Proteins, FluoSpheres, or fluorescent dextran imaging were acquired sequentially (see *Table 2* for tuning). Experiments including dextran were acquired using only two settings: i) high resolution 1024 × 1024 frame size, 0.8 µm Z step size, and 5.2X digital zoom; ii) medium resolution 512 × 32 frame size, 0.5 µm Z step size, and 2.6X digital zoom corresponding to 9 × 144 µm of tissue field. Medium resolution stacks covered the entire ectoderm and endoderm layer of either oral or aboral side. Z-stacks were projected along the y axis (3D projection with X viewing set to 90°) resulting in a pseudo transversal cross section of the tissues (see also *Figure 3—figure supplement 2*). For FluoSpheres, only high resolution z-stacks were acquired.

## Imaging of phalloidin and dextran

Laterally grown microcolonies were incubated for 15 min in S-SFSW supplemented D3K-TRITC. Fixation and decalcification were as before except that decalcification was stopped after 16 hr. Then, the microcolonies grown on their coverslip were rinsed in PBS 3 times and incubated in PBS/3%BSA with Phalloidin-Alexa488 (ThermoFisher A12379) for 4 hr before rinsing in PBS (counterstained with DAPI) and imaged using TCS SP8 inverted microscope with hybrid detectors. Individual channels were acquired sequentially. Three dimensional reconstruction images were computed using the 3D application included in the Leica LasX (Leica) software platform.

**Table 2.** Dye detection.

| Fluo | Laser ex (nm) | Em max (nm) | Detection range (nm) |
|---|---|---|---|
| DAPI | 405 | 461 | 430–470 |
| Symbiodiniaceae* | 405 | | 680–740 |
| GFP* | 488 | | 500–530 |
| Alexa 488 | 488 | 495 | 500–530 |
| TRITC | 543 | 577 | 557–585 |
| TexRed | 543 | 613 | 607–630 |
| FluoSphere | 543 | 605 | 600–640 |

*autofluorescence.

### Automated counting of fluorescent nanoparticles containing vesicles
For the dextran D3K/D10K/D70K pulse chase experiment, high resolution imaging of the different tissues layers were processed with imageJ using the methods developed in *Wang et al. (2014)*. The macro that we used is supplied in the Appendix 1.

### Live imaging of fluorescent dextran
Confocal imaging was performed using TCS SP5 DMI 6000 CS (Leica Microsystems) with a PL FLUO-TAR 16x/0.5 oil objective. For paracellular live imaging experiments, we used microcolonies grown laterally on glass coverslips as in *Muscatine et al. (1997)* and *Venn et al. (2011)*. Briefly, the micro-colony was set in an incubation chamber and analyzed by inverted confocal microscopy from beneath, at the edge of the microcolony where there are gaps in between the growing crystals. Time laps imaging was recorded (one image every 5 s) using appropriate channels. After 3 min 30 s of recording, texasRed-D3K was added to the medium. Videos correspond to screencasts of the LasAM program graphical interface (Leica) displaying the timelaps acquisition (xzyt) in the video mode.

## Acknowledgements

We thank Dominique Desgré for assistance with coral culture. We thank the editorial board and the three anonymous reviewers for their valuable comments which have helped us to improve our manuscript.

## Additional information

### Funding

| Funder | Author |
|---|---|
| Government of the Principality of Monaco | Philippe Ganot<br>Eric Tambutté<br>Natacha Caminiti-Segonds<br>Gaëlle Toullec<br>Denis Allemand<br>Sylvie Tambutté |

This work was supported by the Centre Scientifique de Monaco research program, funded by the Government of the Principality of Monaco.

### Author contributions
Philippe Ganot, Conceptualization, Data curation, Formal analysis, Supervision, Investigation, Visualization, Methodology; Eric Tambutté, Conceptualization, Data curation, Formal analysis, Supervision, Investigation; Natacha Caminiti-Segonds, Formal analysis, Investigation, Methodology; Gaëlle Toullec, Formal analysis, Investigation; Denis Allemand, Resources, Supervision, Funding acquisition;

Sylvie Tambutté, Conceptualization, Resources, Data curation, Supervision, Funding acquisition, Visualization, Methodology, Project administration

#### Author ORCIDs

Philippe Ganot ⓘD https://orcid.org/0000-0003-1743-9709

#### Decision letter and Author response

Decision letter https://doi.org/10.7554/eLife.50022.sa1
Author response https://doi.org/10.7554/eLife.50022.sa2

## Additional files

#### Supplementary files

• Transparent reporting form

#### Data availability

All data generated or analysed during this study are included in the manuscript and supporting files.

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

## Appendix 1

Similar to *Wang et al. (2014)* imageJ macro for counting of the vesicles:
- 1) confocal high resolution Z-stack series were acquired through individual tissue layer.
- 2) individual stacks were processed using imageJ:

*import lif file with/plugins/BioFormats/Bioformats_importer

*Split channels with/Image/Colors/Split_channels

*On the channel of interest, project the Z-stacks corresponding to the tissue of interest with/Image/Stacks/Zprojects (maximum intensity)

*run the following macro with/plugins/macros/

run('8-bit');

run('Subtract Background...', 'rolling = 15');

run('Auto Threshold', 'method = MaxEntropy white');

setOption('BlackBackground', false);

run('Make Binary'); run('Watershed');

run('Analyze Particles...', 'size = 0.1–20 display summarize');

- 3) This imageJ macro gives individual areas in μm$^2$. Given the formula $Area(A) = \pi r^2$, the diameter (D, in nm) of each individual vesicle was deduced in Excel using the formula.

$$D = 2 * \sqrt{(A/\pi)} * 1000$$

- 4) Diameter values for individual Z-stack were processed into frequencies and normalized per 1000 μm$^2$ using excel.

Results are shown in *Figure 7*.

