## [Decision Letter]

**Acceptance summary:**

The cell biology of early metazoans is massively understudied. This work supports the view that the major pinocytic process in anthozoans might be macopinocytosis.

**Decision letter after peer review:**

Thank you for submitting your article "Massive ubiquitous macropinocytosis in anthozoans" for consideration by *eLife*. Your article has been reviewed by three peer reviewers, and the evaluation has been overseen by a Reviewing Editor and Suzanne Pfeffer as the Senior Editor. The reviewers have opted to remain anonymous.

The reviewers have discussed the reviews with one another and the Reviewing Editor has drafted this decision to help you prepare a revised submission.

Even though the manuscript is descriptive, two reviewers with experience in anthozoans agreed that this is a careful study, showing for the first time that basal methazoans take up fluids mainly from the apical side of the oral and aboral cell layers by macropinocytosis. Further, these reviewers felt that this information is of general interest and importance to understand the physiology of this ecologically relevant group and to learn on how they are facing ocean acidification. Based on these comments, it is my pleasure to inform you that the manuscript could be accepted in *eLife* provided you are able to address some of the major concerns to better characterize the endocytic pathway used by corals.

We summarize here the most important issues to be addressed (and at the bottom include the detailed reviews) but we ask that you focus on these summarized comments to guide your next steps.

1) One of the criteria to state that anthozoans take up fluid mainly by macropinocytosis is the use of "specific" inhibitors. There is a general concern related to the claimed specificity of such inhibitors, because, first, their action has not been properly characterized or reported in these organisms, and second, inhibition of their supposed targets might have many other effects besides inhibiting macropinocytosis. In this context, it would be essential to show that the drugs do not compromise the viability of the corals under the experimental conditions used, by showing that their effects are reversible. Further, it would be important to demonstrate that the drugs act on their expected targets. This should not be difficult for inhibitors of actin and tubulin polymerization, where reversible disruption of the cytoskeleton could be demonstrated by fluorescence microscopy. On the other hand, the inhibition with Latrunculin A and EIAPA might affect many other cellular processes, which indirectly impact on endocytic traffic. One of the reviewers, who is an expert in macropinocytosis, suggested the use of PI3K inhibitors to convincingly show that corals use macropinocytosis for fluid uptake. Finally, I think that it would be important to show that the primary endocytic profiles transiently acquired the machinery involved in macropinosome formation (at least actin).

2) A second important criteria used to conclude that anthozoans use macropinocytosis is the size of vesicles accumulating dextran. Besides the technical and conceptual issues raised by two of the reviewers, which strongly suggest to use vesicle volumes and not areas to describe their size, the measures are taken by confocal microscopy earliest upon 15 minutes exposure to the dextran, plus 15 minutes chase. Within this time window, most Transferrin taken up by clathrin-mediated endocytosis in mammals would be found in sorting and recycling endosomes (not in clathrin coated vesicles) and therefore, the timing is not really suitable to define the primary endocytic profiles. Also, as the authors mention, piling up of small vesicles would be scored as big vacuoles with the technique used. Ideally, if the authors want to identify the primary endocytic vesicles derived from the plasma membrane, the experiment should be performed by electron microscopy upon much shorter (5-10 minutes max) exposure to the dextran, followed by a wash out step to remove the surface-exposed dextran. Otherwise, the authors could use a membrane dye such as FM4-64 combined with super-resolution fluorescence microscopy.

3) In the context of defining whether corals macropinocytose constitutively, it would be important to define or clarify if they do so in the absence of rotifer extract. Also, there seems to be an agreement on the lack of conclusion regarding the dextran size sorting issue. I would suggest that you discuss the results in light of the article (Neuhaus et al., 2002); addressing the issue experimentally seems difficult at this point and probably out of the scope of the manuscript. In this context thought, you claim that the delay in the uptake in the D10K and D70K as compared to the D3K in the oral ectoderm is due to the mucus. If you can enzymatically remove the mucus to directly test this hypothesis, the point could be made. Otherwise, please discuss other possibilities such as the existence of different uptake pathways in different cell types.

4) In general terms, all reviewers agreed that the text needs extensive editing (I attach specific comments bellow) and I would certainly encourage you to shorten the Discussion and reduce speculation. The evidence for the existence of other endocytic pathways should be discussed and properly referenced (see specific comments below). The word massive should be removed from the title.

Specific comments from the reviewers:

Reviewer 1:

1) The amount of Materials and methods-like sentences has to be reduced in the Results section. For example, in the subsections “Dynamics of dextran uptake differs between cell layers and size of dextran”, and “Endocytosis occurs at the apical side of the cells”, and many other places.

Reviewer 3:

1) Introduction: Wording a little hard to follow. Thorough editing needed for verb tenses and singular vs. plural.

2) Introduction: second paragraph needs citations throughout.

3) Introduction, third paragraph: should cite primary literature by these authors and others showing coral septate jxn, the Davy review only presupposes their existence (e.g. Barott et al., 2015).

4) Introduction, third paragraph: Barott et al., 2015 also shows evidence of bicarbonate transporters, as well as Na and K ions.

5) Important to mention in the Introduction that the anthozoans studied here have endosymbiotic algae they derive organic nutrients from. These should be referred to as "Symbiodiniaceae" and not zooxanthellae (a nonspecific term that includes many different families of unrelated algae).

6) Just because macropinocytosis is shown here to be utilized by anthozoans for the first time, the phrase 'at the forefront' seems like an overstatement. We know anthozoans do phagocytosis, that's how they take up their symbionts. It also seems safe to assume that have clathrin and caveolae-mediated endocytosis, as many researchers have seen small vesicles that would be consistent with these modes of uptake. Either clarify what macropinocytosis as forefront of or remove.

7) Results should be in past tense.

8) Subsection “Dextran uptake by *Stylophora pistillata* occurs through vesicles”: change 'shows' to 'suggests'.

9) Discussion needs to introduce how closely related these different anthozoans are for readers unfamiliar with cnidaria.

10) Subsection “Macropinosomes and coral calcifying cells”: Other examples of more recent studies showing vesicles in the CE: https://doi.org/10.1371/journal.pone.0209734 and Barott et al., 2015.

11) Subsection “Macropinocytosis and coral health”: wild corals injest plastic microparticles: Rotjan et al., 2019.

12) Subsection “Macropinocytosis and coral health”, second to last sentence: confusingly worded.

13) "fixed on ice" – with what? For how long?

14) Subsection “Incubation of microcolonies with nanoparticles”, last sentence: fixed how? Was it decalcified?

15) It would be helpful to refer to different sized corals with different names. Microcolonies seems more appropriate for the corals growing on coverslips (size not indicated but certainly <2cm) than the 2-3 cm coral fragments hanging from a thread. The coral literature typically refers to corals this size as fragments or nubbins. Minicolonies would be fine as well.

Reviewer #1:

This article reveals some interesting aspects of the massively understudied cell biology of early metazoans. Most of the studies on anthozoans have focused on their symbiotic/mutualistic association with dinoflagellates, but despite their essential place in the Earth ecosystem, little is known about how they communicate with and extract food from their environment. This article is an elegant and thorough, though a bit preliminary, dissection of the major pinocytic process in anthozoans, which appear to be of macropinocytic nature. Overall, the article is very well written and reads like a detective novel, but some parts are overly detailed, with un-necessary technical presentations of otherwise relatively standard procedures (except in the field of anthozoans).

1) The methods used to visualise and describe the endocytic mechanisms and compartments are standard, but applied with extreme care and thoroughness. What is missing is a form of quantification of the progression of the tracers in the various cells and layers. It is likely possible to monitor, quantitate and represent the (differential, see point 2 below) advance of the tracers in the form of a "gradient" through the tissue, similarly as is performed when researchers measure and model the gradients of morphogens progressing through a tissue such as the *Drosophila* imaginal disk.

2) The authors present some interesting data about "size selection" of two different dextran molecules used as fluid-phase tracer. First, such a phenomenon has been reported earlier (for example in Neuhaus et al., 2002) and the authors could compare/contrast their findings with those. Second, because the "molecular sieving" probably does not occur during the ingestion, subsequent macropinosome fragmentation, tubulation and cargo sorting, or other size-discriminating phenomena have to take place. The authors could try to visualise such subsequent phenomena by live microscopy, measuring the fluorescence ratio as a read-out of the selection.

3) Again, the manuscript is well written, but the amount of Materials and methods-like sentences has to be reduced in the Results section. For example, in the subsections “Dynamics of dextran uptake differs between cell layers and size of dextran” and “Endocytosis occurs at the apical side of the cells”, and many other places.

4) The macropinocytic process revealed in this study, as well as the progression through tissue layers are well documented (but see point 1-). Now, in terms of mechanistic insight, beside the size of the compartments, which classify plausibly the process as macropinocytic, the authors should relativise their conclusions based on the use of "specific" drugs to block the pathway, because this is based on experiments performed in evolutionarily distant organisms and the molecular targets and mechanisms of action of the drugs might not be conserved.

5) The authors speculate about the fact that fluid-phase tracers are taken up in a "constant" manner and might become concentrated during the uptake process (subsection “Macropinocytosis and signaling”). The former sounds plausible, but there is no real proof of that. The latter can be substantiated by an experiment (it is easy to measure the intensity of fluorescence per volume units during the process), or the authors have to cut short their speculations, especially "It is thus tantalizing to extrapolate a role for macropinocytosis in cell to cell signaling pathways". Keep this to write a review once the article is published.

Reviewer #2:

This paper potentially adds any interesting piece of biology to the macropinocytosis field, although a lot of it is also very descriptive for *eLife*. The core is the observation that particles that are too big to be taken up through transporters are being endocytosed via macropinocytosis. That point is clearly made and may be very important for the field. The finding that the macropinocytosis comes from a particular surface seems clear and may be important – a reviewer with expertise in anthozoa will have to comment.

Others are less sound, for one reason or another:

- The focus on the different sizes of dextrans is currently unfocused – there are no conclusions drawn, and "the reason for this is unknown". This party of the story is not ready for *eLife* yet – it's a set of results that don't yet lead to understanding. Interestingly, the authors may find some clues in the work of Thierry Soldati, who has published on the differential sorting of different dextran sizes. I think the relevant paper was about eight years ago, but can't remember precisely.

I would like to see this work led to a justified scientific conclusion.

- The inhibitors are called "specific" in the Abstract (it is incidentally odd to say that without saying what they're specific for) – but they aren't. Blocking Nhe1 for 45' could really harm cells; the authors need to show that the rest of cellular physiology is unaffected and that the organisms can recover. The latrunculin is possibly worse – single cells that have been incubated that long in high concentrations are slaughtered; it blocks single-cell macropinocytosis assays within minutes, but also blocks other vesicle traffic, makes the membranes go floppy and blebby, and more generally breaks down cell architecture over tens of minutes. It's conceivable that it takes that long for the drug to diffuse in to the cells of interest (have the authors tested this?) but in any case they should consider secondary effects from actin disruption, including but not only changes to cell viability.

I would like to see careful controls that these inhibitors were behaving specifically, and the appropriate part of the Abstract rewritten. Also, why not examine π 3-kinase inhibitors? Those are (somewhat) less likely to kill the cells.

Measuring vesicles and expressing the size as an "area" is not sensible. Vesicles are spherical; the images presumably slice through them at different parts; this needs to be considered and turned into an informed estimate of a volume. And saying " size of the vesicles that we have observed.… is between 350 to 1500 nm" is also unclear – does it mean "the diameter of the slices in EM" or "the diameter of slices in the confocal" or is it a measure of the true diameter like widefield?

The references for dextran in imaging are very late and miss the real innovators – suggest citing for example Clarke et al., 2002.

Reviewer #3:

This study describes for the first time a mechanism, macropinocytosis, for the uptake and intracellular transport of dissolved compounds and nanoparticles within corals and two other anthozoans. The use of fluorescence and electron microscopy, coupled with pharmacological inhibitors of macropinocytosis, provide convincing support for this mechanism. It would be nice to see F-actin staining of the tissues across the pulse/chase time series to help confirm macropinocytosis (and that the 'ruffles' are not just cilia and/or microvilli), something that given the experience of these authors would not be hard to do. Directional transport from the apical side of both endo- and ectodermal cells is convincingly demonstrated for the oral tissues, however I am not convinced they have shown this pattern of uptake in the calicoblastic epithelium. Instead, it looks like the dextrans are moving through the aboral endodermal cells (taken up apically as indicated) and then transported throughout those cells and across via the basolateral side of the CE cells. I was also surprised to read in the Materials and methods that the corals are given a rotifer extract (feeding stimulant) during all of the incubations, but this is never mentioned in the Discussion. This is an important point to bring up and I would like to know if the authors first tried their experiments without the extract and saw no/less macropinocytosis activity.

Overall this study is an important step forward in our understanding of coral biology that has important implications for cnidarian physiology, particularly as it relates to coral exposure to water soluble pollutants but also for regulation of coral uptake of inorganic and organic nutrients. The authors also argue that corals may be a useful model system for the study of macropinocytosis as it relates to human health and drug delivery, a valuable point that was not articulated as clearly as it should be. Overall, it does not seem surprising that anthozoans use macropinocytosis given that this mechanism is conserved with the most basal eukaryotes, and the authors overstate this novelty. That said, to have described this mechanism for the first time in a class of basal metazoans is an important and interesting advance in our understanding of the physiology of this ecologically and evolutionarily significant group of taxa. It also is a step towards describing the mechanisms of intra- and inter-cellular transport necessary for coral calcification, a timely topic of study for understanding how corals facing ocean acidification may fare.

The text of the manuscript needs to be thoroughly edited for flow and clarity.

[Editors' note: further revisions were suggested prior to acceptance, as described below.]

Congratulations, we are pleased to inform you that your article, "Ubiquitous macropinocytosis in anthozoans", has been accepted for publication in *eLife*. However, before full acceptance you would need to address a few points by rewriting some sections:

This article reveals some interesting aspects of the massively understudied cell biology of early metazoans and supports the view that the major pinocytic process in anthozoans might be macopinocytosis. There is a general agreement that you have done a major effort:

1) To demonstrate that the Latrunculin A effect is reversible;

2) To image actin in association with the dextran loaded structures;

3) To demonstrate that inhibition of PI3K inhibit dextran uptake; and,

4) To more properly quantify the vesicle-vacuole size.

The reviewers also agree though that the new results further support the view that macropinocytosis might be the prevalent endocytic pathway in corals but they do not conclusively prove it because:

1) The actin associated with the dextran vesicles-vacuoles might still correspond to endosomal actin rather than actin structures associated with forming macropinosomes; and,

2) Because other endocytic mechanisms such as the GEEC pathway are also sensitive to PI3K inhibitors, EIPA and Latrunculin A. In any case, since precise dissection of the endocytic pathways in corals is out of the scope of the manuscript and experimentally very challenging, we are willing to accept the manuscript with no further experimental work, but we ask you to please discuss the fact that other endocytic mechanisms such as the GEEC pathway cannot be excluded.

Some other points need to be addressed in addition:

1) The reviewers accept that you did not really probe that the drugs target the expected proteins, but discuss instead the likelihood that this is the case in evolutionary terms. This is fine for the experiments where you observe an effect of the drug. However, it would be important to show that colchicine indeed depolymerizes the microtubules in your system, since you claim that microtubule depolymerization does not affect dextran uptake. If you cannot do this, this negative result should be omitted.

2) All reviewers also agree that the Discussion is still too long and re-describes some of the results in more detail than needed. The comparison to early TEM work in corals showing the presence of vesicles seems particularly long, as is the discussion of the possible importance of coral macropinocytosis with regard to microplastics. Those last three sections could be combined and condensed into a 'Macropinocytosis and coral physiology' section. Please condense the Discussion to no more than 5 pages double spaced in your draft manuscript.

---

## [Author Response]

Several very important concerns were raised and complementary experiments were required in order to improve our demonstration of macropinocytosis in the anthozoan coral *Stylophora pistillata* (Cnidaria). These major points included i) the use of more specific inhibitors such as PI3K inhibitors, ii) showing that the effect of the inhibitors were reversible, and iii) trying to visualize the interplay between the movement of the plasma membrane and/or iv) the cortical actin network associated with macropinocytosis. Indeed, in other eukaryotic models such as vertebrate cells or the unicellular protist *Dictyostelium* where the mechanisms of macropinocytosis have been carefully studied, plasma membrane ruffling guided by F-actin reorganization was demonstrated and could be inhibited.

We have tried our best to comply with these very justified demands using different experimental set-ups and we believe that we managed to fill many of the gaps raised by our initial submission. However, working with a whole organism, moreover a coral, is certainly not as straightforward as working on cells (or even other organisms), and some of our experimental attempts were unsuccessful, likely due to the complexity associated with coral biology.

A) General answers and major modifications

1) Membrane visualization: we tried to visualize the cellular membranes of the different tissues using 3 different membrane dyes, i.e. FM4-64 (thermofisher), lipilight (idylle) and Vybrant DiO (thermofisher), without success. Whether using live or fixed samples, we never managed to label any plasma or intracellular membrane. The reason of this is unknown, and would probably require further investigation that falls outside the frame of our present manuscript. We will explore this intriguing “non-result” in the future.

2) Actin visualization: on the one hand, Sir-actin probes (Cytoskeleton) on live animals did not work. On the other hand, we used fluorescent-conjugated Phalloidin on fixed samples to visualize the F-actin network after a short pulse (15 minutes) of dextran, and obtained pretty good results. We had to modify the experimental set-up in the sense that we used laterally grown coral colonies instead of coral fragments: laterally grown coral colonies are grown on coverslips producing a rather thin layer of skeleton. After fixation, they can be (partially) decalcified in a relatively short period of time, i.e. over-night instead of 3-5 days in the regular protocol. Hence, we experienced a much better preservation of the F-Actin network. Also, we moved to a more sensitive confocal equipped with hybrid detectors (Leica SP8). *In fine*, we were able to clearly image the actin sheet associated with the forming macropinosome (Dextran labeling). Using 3D reconstructions (Leica LasX), we were able to visualize a strong signal of actin (which we called a “cap”) associated with the majority of dextran vesicles. This is now Figure 10 and Videos 3-5 of the manuscript.

3) PI3K inhibitors: we first used LY294002 (cayman chemical) as it is a reversible PI3K inhibitor. However, this drug was inadequate as it had another very strong and unexpected side effect on the general autofluorescence of the cells. We had never observed such a phenotype previously. Author response image 1 shows an example of imaging after LY294002 treatment. We also included a phylogenetic analysis the PI3K family in *Stylophora*, showing that there is only one PI3K homolog, against 4 in human.

**Author response image 1. respfig1:** LY294002 treatment triggers unexpected secondary effect and phylogenetic comparison of the PI3K family in Stylophora pistillata vs human. (**A and B**) Confocal Z-stack of *Stylophora* incubated in LY294002. Red=Dapi; Green=dextra; gray=Zoox, and Cyan is the GFP chanel. A and B correspond to the oral ectoderm and the aboral endoderm, respectively. Note that in the oral ectoderm, the cell layer is filled with GFP-like signal that is never seen under other conditions. In B, the GFP-like signal is less strong (although different than controls), and the vesicles of dextran are abnormally large (4-5 μm in diameter). However, based on the side effect of LY294002, this assay was not validated. (**C**) PhyML phylogeny of the PI3K families in human (names in black) and in *Stylophora* (names in blue). In *Stylophora*, there is only one cognate PI3K homolog (4 in human). The other sequences are PI3K-related members (i.e. PI3K-C2, PI3K-type3 and PI4K).

On the other hand, we successfully used Wortmannin (Selleckchem) another PI3K inhibitor, although irreversible. Like EIPA, Wortmannin blocked Dextran uptake, showing that macropinocytosis in anthozoans is PI3K dependent as in other eukaryotes. This is now Figure 8.

4) Inhibitor reversibility: in the first version of our manuscript, we had shown that the actin inhibitor Latrunculin inhibited dextran uptake at 500 nM (not effective on the aboral endoderm) and 1000 nM (effective on the aboral endoderm). In order to verify that latrunculin (LatA) was not destroying the cellular integrity of the animals’ tissues (which would also end up as an apparent inhibition), we tested reversibility. Corals are colonial organisms. This means that a fragment (corresponding to approx. 100 individuals or polyps) can be cut into 2 pieces without much harm to the colony. In order to assess the reversibility of the drug, we used a similar set-up than in the first version, albeit this time, after the LatA inhibition and dextran pulse, we sampled only one half of the colony. The second half was then submitted to a second step with another dextran pulse after a 4 hour wash without the drug. At 500 nM LatA, we again experienced a strong impairment of the dextran uptake (30 minutes pulse), except in the aboral endoderm (step1). However, after a 4 hours wash in regular seawater, the same colony had recovered the ability to uptake dextran (step2). Concurrently, in the aboral endoderm (as well as in the oral cell layers), inhibition of dextran uptake was achieved at 1000 nM, and was reversibly restored after washing. However, at this concentration, we experienced that the cells composing the thin calicoblastic cell layer were rounded and not forming a true epithelium. Nevertheless, we conclusively showed that latrunculin specifically (since reversible) blocked actin in the process of macropinocytosis, with tissue layers responding differently depending on drug dosage. The 500 nM experiment is now Figure 8 (and figure supplements).

We believe that within the frame of a technically challenging investigation of cell biology in corals, this novel set of experiments should provide sufficient additional proof to conclude that macropinocytosis is indeed the process that we show in the present manuscript. We have developed several innovative imaging approaches for the field of coral biology that we hope should satisfy the general and specific demands of the reviewers, as summed up by the editor.

Other figure changes:

i) Figure supplements have to complement a main text figure. We created a new Figure 1 (which was previously in our supplementary data) to schematize the anatomy and histology of the coral. This helps the comprehension of the following figures for scientist non-familiar to scleractinian corals. Consequently, all previous figures have their number incremented by +1 (ex-Figure 1 is now Figure 2 etc.)

ii) We totally agree that describing vesicles with surface areas in µm^2^ was inappropriate. We went back to the measurement files and converted every area (which corresponds to Z-stack projections) into diameters (A=πr^2^). We then reconstructed the distributions of the different diameter ranges, using 200 nm steps. Figure 7 shows vesicles diameters in nm. There was no change to the conclusions of this part.

iii) We changed one picture in the Figure 2—figure supplement 1C which was showing transmission electronic microscope picture of the aboral endoderm. The previous picture was misleading with regard to extracellular microvilli versus intracellular macropinosomes. The new picture clearly shows that endodermal cells contain (forming) macropinosomes (mp) as part of the apical plasma membrane, like in the ectoderm (b). Microvilli (Mv) are now indicated with arrows to avoid confusion. To our understanding, the positioning of these vesicles is not likely to represent large late endosomes resulting from the fusion of smaller early endosomes. The text describing the macropinosome formation as part of the apical membrane has been changed accordingly.

B) Detailed answers to the issues summarized by the editors

[…] We summarize here the most important issues to be addressed (and at the bottom include the detailed reviews) but we ask that you focus on these summarized comments to guide your next steps.1) One of the criteria to state that anthozoans take up fluid mainly by macropinocytosis is the use of "specific" inhibitors. There is a general concern related to the claimed specificity of such inhibitors, because, first, their action has not been properly characterized or reported in these organisms, and second, inhibition of their supposed targets might have many other effects besides inhibiting macropinocytosis. In this context, it would be essential to show that the drugs do not compromise the viability of the corals under the experimental conditions used, by showing that their effects are reversible. Further, it would be important to demonstrate that the drugs act on their expected targets. This should not be difficult for inhibitors of actin and tubulin polymerization, where reversible disruption of the cytoskeleton could be demonstrated by fluorescence microscopy. On the other hand, the inhibition with Latrunculin A and EIAPA might affect many other cellular processes, which indirectly impact on endocytic traffic. One of the reviewers, who is an expert in macropinocytosis, suggested the use of PI3K inhibitors to convincingly show that corals use macropinocytosis for fluid uptake. Finally, I think that it would be important to show that the primary endocytic profiles transiently acquired the machinery involved in macropinosome formation (at least actin).

See answers in A.

2) A second important criteria used to conclude that anthozoans use macropinocytosis is the size of vesicles accumulating dextran. Besides the technical and conceptual issues raised by two of the reviewers, which strongly suggest to use vesicle volumes and not areas to describe their size, the measures are taken by confocal microscopy earliest upon 15 minutes exposure to the dextran, plus 15 minutes chase. Within this time window, most Transferrin taken up by clathrin-mediated endocytosis in mammals would be found in sorting and recycling endosomes (not in clathrin coated vesicles) and therefore, the timing is not really suitable to define the primary endocytic profiles. Also, as the authors mention, piling up of small vesicles would be scored as big vacuoles with the technique used. Ideally, if the authors want to identify the primary endocytic vesicles derived from the plasma membrane, the experiment should be performed by electron microscopy upon much shorter (5-10 minutes max) exposure to the dextran, followed by a wash out step to remove the surface-exposed dextran. Otherwise, the authors could use a membrane dye such as FM4-64 combined with super-resolution fluorescence microscopy.

We think that there was a misunderstanding with the timing of our dextran incubations (pulses). With the coral *Stylophora pistillata*, we have been incubating (without chase/wash) the fragments for as short as 5 minutes, as shown in the Figure 3—figure supplement 3 (even shorter pulses were tested; not shown). However, it is only after 10 minutes that the first vesicles were apparent in the aboral endoderm (Figure 3—figure supplement 3) or 15 minutes that they became apparent in most cells (Figures 4A, 5B, 10). Several parameters have to be considered for the dextran to arrive from the surrounding seawater to the 4 cell layers: for the oral ectoderm (in contact with the seawater), the mucus layer is a barrier. For the oral and aboral endoderms, the time for the renewal of the coelenteric cavity, which is the time for the dextran to first arrive in contact with the cellular membranes, is within the 5 minute range (discussed in the Discussion paragraph “Particle diffusion and cell progression of macropinosomes”). Of note, detection of a forming macropinosome requires a sufficient concentration of Dextran to have a detectable signal. Importantly, when we injected Dextran directly into the coelenteric cavity of the sea anemone *Anemonia viridis* (Figure 5—figure supplement 2), the first vesicles could be seen after only 1 minute, and were definitively evident after 4 minutes. Finally, in the calicoblastic ectoderm facing the skeleton, the dextran present in the coelenteric cavity needs to cross two septate junctions and the mesoglea before reaching the cells’ apical membranes, an additional time lag that we estimate to represent approx. 5 minutes. Thus, the time required for forming large (Diam. >350 nm) vesicles can be estimated to be less than 4 minutes. This is in agreement with experiments demonstrating macropinocytosis on cell lines (in direct contact with their media) from other organisms. It is less compatible with the accumulation of dextran in late endosomes.

3) In the context of defining whether corals macropinocytose constitutively, it would be important to define or clarify if they do so in the absence of rotifer extract. Also, there seems to be an agreement on the lack of conclusion regarding the dextran size sorting issue. I would suggest that you discuss the results in light of the article (Neuhaus et al., 2002); addressing the issue experimentally seems difficult at this point and probably out of the scope of the manuscript. In this context thought, you claim that the delay in the uptake in the D10K and D70K as compared to the D3K in the oral ectoderm is due to the mucus. If you can enzymatically remove the mucus to directly test this hypothesis, the point could be made. Otherwise, please discuss other possibilities such as the existence of different uptake pathways in different cell types.

The possibility of “inducing” macropinocytosis by supplying rotifer extract with the different tracers is a relevant point that we eluded in the first version of the manuscript. In the beginning of our investigation, we were using Sterile Filtered Seawater only to incubate the fragments. And we indeed observed macropinocytosis. Then, in order to avoid potential nutritional stress, we decided to conduct all our experiments with rotifer extracts, as there was no apparent changes in our observations. Thus, although we did not quantify the potential effect of the supplemented extracts, we can rule out the possibility that macropinocytosis is induced by rotifers. We added a comment in the relevant Materials and methods section.

With regard to the “selectivity” of the dextran size in the different tissues, we would rather prefer to avoid drawing definitive conclusions. It is an intriguing phenomenon that we observed (Figures 3 and 4) and measured (Figure 7) especially in the oral ectoderm. Several hypotheses could be drawn. The first and most easily accepted is size selectivity due to the mucus layer. A clear stack of Dextran is visible accumulating only after 5 min on the apical side of the oral ectoderm, where the mucus layer lies (see figures). This Surface Mucopolysaccharide Layer (mucus) is particularly abundant, representing a biological mesh and is part of the mechanisms for heterotrophic feeding from Dissolved Organic Material (DOM) to trapped bacteria (see review by Brown and Bythell, 2005 for extensive descriptions). Another possibility for nanoparticles selectivity could be the net charge of the particles (Table 2). The combination of a mucopolysaccharidic mesh layer with differently charged particles should also be taken into account. However, we certainly cannot exclude additional endocytic mechanisms that would contribute to the selective uptake of nanoparticles as was nicely demonstrated in *Dictyostelium* (Neuhaus et al., 2002). In this latter scenario, a “classical” endocytic pathway (e.g. clathrin-mediated) would selectively endocytose D3K in small vesicles that would later fuse as large late endosomes or with macropinosomes. This could explain why, after 15 minutes pulse and 4 hours chase (wash), there are still discrepancies between the different dextran size uptakes. However, short pulse experiments (e.g. Figure 4A) always show a rapid preferential uptake of D3K versus D10K, which would be better explained by mucus selective retention. Since we have yet no experimental evidence to conclude on the matter, we have modified the text to leave the field open and discussed the aforementioned hypotheses.

4) In general terms, all reviewers agreed that the text needs extensive editing (I attach specific comments bellow) and I would certainly encourage you to shorten the Discussion and reduce speculation. The evidence for the existence of other endocytic pathways should be discussed and properly referenced (see specific comments below). The word massive should be removed from the title.

We have reduced the Discussion and removed the speculative part on cell-cell signaling via macropinocytosis. The 15 following points (see below) were modified as requested. The entire manuscript has been revised by a professional text editor. We hope that this new version, which includes multiple text changes, will now be eligible for publication.

C) Specific answers to the reviewers

Reviewer #1:1) The methods used to visualise and describe the endocytic mechanisms and compartments are standard, but applied with extreme care and thoroughness. What is missing is a form of quantification of the progression of the tracers in the various cells and layers. It is likely possible to monitor, quantitate and represent the (differential, see point 2 below) advance of the tracers in the form of a "gradient" through the tissue, similarly as is performed when researchers measure and model the gradients of morphogens progressing through a tissue such as the *Drosophila* imaginal disk.2) The authors present some interesting data about "size selection" of two different dextran molecules used as fluid-phase tracer. First, such a phenomenon has been reported earlier (for example in Neuhaus et al., 2002) and the authors could compare/contrast their findings with those. Second, because the "molecular sieving" probably does not occur during the ingestion, subsequent macropinosome fragmentation, tubulation and cargo sorting, or other size-discriminating phenomena have to take place. The authors could try to visualise such subsequent phenomena by live microscopy, measuring the fluorescence ratio as a read-out of the selection.

Thank you for these two very constructive comments. We have the feeling however that they fall beyond the scope of the present study which is the journey of nanoparticles from the seawater to the cell uptake. Nevertheless, the actual study shall be continued; with the use of more advanced imaging facilities as well as combination of other tracers, in order to tentatively decipher the nanoparticle selectivity (size/charge/composition) as well as the cellular fate of the macropinosomes inside the cell. And live imaging using laterally grown micro-colonies is, among others, some of the tools we are currently developing.

3) Again, the manuscript is well written, but the amount of Materials and methods-like sentences has to be reduced in the Results section. For example, in the subsections “Dynamics of dextran uptake differs between cell layers and size of dextran” and “Endocytosis occurs at the apical side of the cells”, and many other places.

See section B for answer.

4) The macropinocytic process revealed in this study, as well as the progression through tissue layers are well documented (but see point 1-). Now, in terms of mechanistic insight, beside the size of the compartments, which classify plausibly the process as macropinocytic, the authors should relativise their conclusions based on the use of "specific" drugs to block the pathway, because this is based on experiments performed in evolutionarily distant organisms and the molecular targets and mechanisms of action of the drugs might not be conserved.

Macropinocytosis has been studied in eukaryotes as distant as the amoebae *Dictyostelium* and vertebrates using similar inhibitors. Cnidaria is the sister group to Bilateria, part of the Epitheliozoa. Moreover, the genomic sequencing efforts have shown that in most cases, gene sequences similarity was higher between cnidarian and vertebrate than between protostomian (e.g. *Drosophila, C. elegans*) and vertebrates. Actin is such a conserved molecule that there is no doubt about the inhibitory effects of Latrunculin. EIPA has been shown to be specific to NHE in *Stylophora* (Laurent et al., 2014). Although LY294002 had unexpected side effects that made its use invalid, we now show the effect of Wortmannin: the phylogenetic tree of the PI3K shows that the *Stylophora* homolog is positioned between the human homologs, which reflect the high level of sequence homology between the human and cnidarian sequences. Hence, we are confident that the action of the drugs used in our study is specific.

5) The authors speculate about the fact that fluid-phase tracers are taken up in a "constant" manner and might become concentrated during the uptake process (subsection “Macropinocytosis and signaling”). The former sounds plausible, but there is no real proof of that. The latter can be substantiated by an experiment (it is easy to measure the intensity of fluorescence per volume units during the process), or the authors have to cut short their speculations, especially "It is thus tantalizing to extrapolate a role for macropinocytosis in cell to cell signaling pathways". Keep this to write a review once the article is published.We have removed this paragraph as it was too speculative.Reviewer #2 and Reviewer #3

See sections A and B for responses.

[Editors' note: further revisions were suggested prior to acceptance, as described below.]

[…] Some other points need to be addressed in addition:1) The reviewers accept that you did not really probe that the drugs target the expected proteins, but discuss instead the likelihood that this is the case in evolutionary terms. This is fine for the experiments where you observe an effect of the drug. However, it would be important to show that colchicine indeed depolymerizes the microtubules in your system, since you claim that microtubule depolymerization does not affect dextran uptake. If you cannot do this, this negative result should be omitted.

We removed the (negative) results from the colchicine experiment, i.e. the corresponding Figure 9—figure supplement 2 and related sentences in the Materials and methods, Results and figure legends sections.

2) All reviewers also agree that the Discussion is still way too long and re-describes some of the results in more detail than needed. The comparison to early TEM work in corals showing the presence of vesicles seems particularly long, as is the discussion of the possible importance of coral macropinocytosis with regard to microplastics. Those last three sections could be combined and condensed into a 'Macropinocytosis and coral physiology' section. Please condense the Discussion to no more than 5 pages double spaced in your draft manuscript.

We modified the Discussion section:

– We added a small paragraph on the CLIC/GEEC pathways and its putative participation in the formation of the vesicles that we observed.

– We shortened the Discussion down to 5 pages, which includes removing detailed re-description of some of the results and merging of the previous last 3 sections into one 'Macropinocytosis and coral physiology' section.

– We added an acknowledgment paragraph.